# Perdigão 2015: methodology for atmospheric multi-Doppler lidar experiments

Nikola Vasiljević[1], José M.L.M. Palma[2], Nikolas Angelou[1], José Carlos Matos[3], Robert Menke[1], Guillaume Lea[1], Jakob Mann[1], Michael Courtney[1], Luis Frölen Ribeiro[4], and Vitor M.M.G.C. Gomes[2]

[1]Technical University of Denmark - DTU Wind Energy, Frederiksborgvej 399, Building 118-VEA, 4000 Roskilde, Denmark
[2]Faculty of Engineering of the University of Porto, Rua Dr. Roberto Frias, 4200-465 Porto, Portugal
[3]Institute of Science and Innovation in Mechanical and Industrial Engineering, Rua Dr. Roberto Frias, 4200-465 Porto, Portugal
[4]Polytechnical Institute of Bragança, Campus de Santa Apolónia, 5300-253 Bragança, Portugal

*Correspondence to:* Nikola Vasiljević (niva@dtu.dk)

**Abstract.** The long-range and short-range WindScanner systems, multi-Doppler lidar instruments, can map the turbulent flow around a wind turbine and at the same time measure mean flow conditions over an entire region such as a wind farm. As the WindScanner technology is novel, performing field campaigns with the WindScanner systems requires a methodology that will maximize the benefits of conducting WindScanner-based experiments. Such a methodology is presented and discussed through its application in a pilot experiment that took place in a complex and forested site in Portugal where for the first time the two WindScanner systems operated simultaneously. The methodology consists of 10 steps. The steps are presented and the implementation of each step is demonstrated. Overall, the demonstration of the methodology in the pilot experiment resulted in a detailed site selection criteria, well-thought-out experiment layout, novel flow mapping methods and high quality flow observations, all of which are presented in this paper.

## 1 Introduction

In wind energy research, field experiments are important for wind resource evaluation but also to establish, validate and improve theories and wind flow models. If experiments are well planned, designed, executed and reported, the field datasets have a long lifetime and are a firm basis for the advancement of our knowledge on atmospheric flows.

A large number of field experiments addressing flows over hills (cf., Taylor et al., 1987) were carried out between 1979 and 1986. These field experiments provided the experimental validation of the models of the wind industry resource assessment (Jackson and Hunt, 1975; Mortensen et al., 2004; Walmsley et al., 1986). Among the field experiments of the past, the Askervein hill experiment (Taylor and Teunissen, 1987; Walmsley and Taylor, 1996) is an example of a field campaign dataset that has been in use for more than 30 years. The increased computational power and developments in both numerical techniques and implementation of the flow physics have enabled the use of computational fluid dynamics and mesoscale models at a larger scale, which calls for validation with new field experiments. The need for new and higher quality field datasets, recognized

in various wind energy related forums (e.g., EWEA, 2005, 2008; Shaw et al., 2009; van Kuik et al., 2016), is driven by the increasing size of the modern wind turbines often located in complex orographies.

Due to the costs of tall meteorological masts, especially in complex terrain, it is unrealistic to sample the wind within an entire region occupied by today's largest wind turbines or farms with traditional anemometry. This is exactly what can be achieved with multi-Doppler lidar systems. The reason for using multiple lidars is that a single lidar can directly sense only line-of-sight (LOS) or radial wind speed, which is a projection of the wind vector on a laser light propagation path (see Cariou and Boquet, 2011). Indirectly, by employing single-Doppler retrieval techniques (e.g., Browning and Wexler, 1968; Strauch et al., 1987) on a number of LOS measurements, single lidars can retrieve an accurate wind vector information if the flow is horizontally homogeneous (see Courtney et al., 2008; Peña Diaz et al., 2009) . In complex terrain, the flow rarely satisfies this assumption, which leads to erroneous retrievals of the wind vector by single lidars (Bingöl et al., 2009).

Employing at least two lidars and intersecting their beams at a point of interest is required to directly measure two components of the wind at that point, while to fully characterize the wind vector at least three laser beams should intersect at a given point. By moving the beam intersection over an area or volume of interest, the flow can be mapped and resolved in two or all three dimensions. Dual-Doppler setups consisting of two scanning lidars have been used in several large atmospheric studies (e.g., McCarthy et al., 1982; Newsom et al., 2005; Collier et al., 2005; Grubišić et al., 2008), while their application in wind energy related studies is rather recent (e.g., Iungo et al., 2013; Newsom et al., 2015) . Triple-lidars were primarily used to demonstrate the capability of accurately retrieving the wind vector in a single point (Mann et al., 2009) or in multiple points distributed along single vertical axis (see virtual masts in Fuertes et al., 2014; Newman et al., 2016; Lundquist et al., 2017).

As discussed in Vasiljević et al. (2016a), lidars in the previously mentioned studies were individually configured, run and monitored, thus there was no central computer that would ensure their synchronicity. Also, commercially available lidars were used in the majority of these studies, and the configuration flexibility and available auxiliary information of such devices is usually limited (e.g., no accurate timing of scans, only simple scanning methods available).

Efforts put into the WindScanner.dk project have resulted in two specific and highly configurable multi-Doppler lidar instruments, known as the long- and short- range WindScanner systems (Mikkelsen, 2014), specifically designed to address the aforementioned needs of wind energy research for new measurements. These instruments unlocked the potentials not only to measure the wind field at a single point located within the rotor swept area, but also to map entire wind fields within the volume occupied by a modern wind turbine and wind farms. As the WindScanner systems are a novel and extremely flexible wind measurement technology there is an increased chance of misconducting experiments and not producing the best possible results. In order to maximize the benefits of using the WindScanner technology, there is a need to establish a methodology for WindScanner based atmospheric experiments.

In this paper, we propose such a methodology. The methodology will be discussed through its application to a pilot experiment that took place in a complex and forested site in Portugal. The paper is organized as follows: Section 2 describes the WindScanner systems, Section 3 and Section 4 present the methodology and a detailed description of how it was applied, the results of the methodology demonstration and future work are presented in Section 5, while Section 6 provides the concluding remarks.

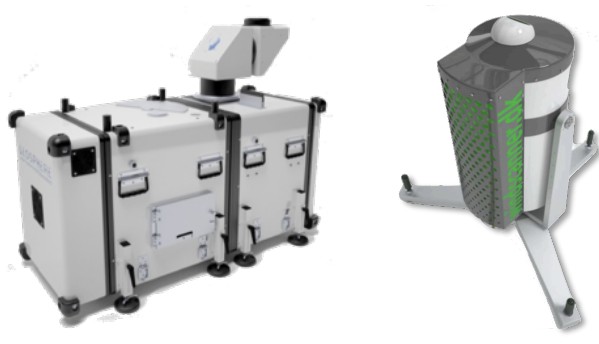

**Figure 1.** DTU Wind Energy scanning lidars. Long-range WindScanner (left); Short-range WindScanner (right)

## 2 WindScanner systems

A WindScanner system consists of two or more spatially separated scanning lidars (long- or short- range WindScanners, Figure 1) that are coordinated or controlled by a master computer. The first generation of the long-range WindScanner (LRWS) and short-range WindScanner (SRWS) originate from commercially available vertical profiling lidars Windcube 200 and ZephIR, respectively. Under the WindScanner.dk project, DTU Wind Energy, with the expert support from specialized industrial partners, developed scanner heads, converting the profiling lidars into the scanning lidars that make up the LRWS and SRWS. In addition to the hardware developments, DTU Wind Energy developed specific software solutions for both WindScanner systems (e.g., Vasiljević et al., 2016a).

The WindScanners have been specifically tailored to perform user-defined and time-controlled scanning trajectories, known as complex trajectories, either independently or in a synchronized mode. The lidars in the long-range WindScanner system are usually connected to the master computer using a 3G network (Vasiljević et al., 2016b). The short-range WindScanner system is formed by connecting the master computer with the lidars via a 300-meter long optic fiber, controlling them using a MACRO (Motion and Control Ring Optical) digital interface (see Delta Tau).

SRWSs provide high-frequency measurements (up to 400 Hz) of the flow while probing the atmosphere with a relatively small probe length. This allows resolving small length scales (down to a few cm) and short time scales (down to 2.5 ms) of the flow. As SRWSs are based on the CW lidar technology there are several limitations. Since the LOS speed at a given point is resolved by focusing the laser beam, the probe length evolves with range. This limits the maximum range of a SRWS to about 150 m. Furthermore, since the beam can be focused only at a single point in time, thus radial velocity from one single range can be resolved. However, a high measurement rate compensates this limitation in ranging. Based on these characteristics the short-range WindScanner system is ideal for mapping of turbulent flow features around a single wind turbine rotor (e.g., Immas et al., 2015) or around a small scale orographic feature (e.g., Lange et al., 2016).

LRWSs have a larger probe length (minimum 25 m) and lower measurement frequency (10 Hz at best, typically 1 Hz) than SRWSs. Since LRWSs are based on the pulsed technology their probe length is constant with range. Furthermore, LRWSs can simultaneously retrieve radial velocity from a number of ranges along the laser light propagation path. This number is

**Table 1.** WindScanners' characteristics overview

|  | Long-range WindScanner | Short-range WindScanner |
|---|---|---|
| Technology | Pulsed | Continuous wave (CW) |
| Range (m) | 50 to 8000 | 10 to 150 |
| Maximum measurement rate (Hz) | 10 | 400 |
| Range gates per LOS | up to 500 | 1 |
| Probe length (m) | 25, 35 or 70 (fixed with range) | 0.2 to 40 (evolving with range) |
| Scanner head | Triple-mirror based gear-box driven | Double-prism based belt driven |
| Atmospheric coverage | Hemisphere | $120^{\circ}$ cone |
| Weight (kg) | 180 | 300 |
| Peak power consumption (kW) | 1.7 | 2.4 |
| Environment protection | IP65 | IP62 |
| Temperature range ($^{\circ}$C) | -10 to 40 | -10 to 40 |

limited by the computational power of the lidar. This particular characteristic of ranging compensate for a lower measurement
frequency since at any given measurement rate LRWS can provide a 'snapshot' of the atmosphere up to several kilometers
along a single LOS. The maximum range of LRWSs is about 8 km, which has been typically observed in offshore conditions
(Floors et al., 2016). Due to its characteristics, the long-range WindScanner system is primarily intended for measurements of
mean flow fields within a large area of the atmosphere (e.g., Berg et al., 2015).

A summary of the characteristics of the two types of WindScanners is given in Table 1. A more detailed description of the
long-range WindScanner system is provided in Vasiljević (2014b); Vasiljević et al. (2016a), while Mikkelsen et al. (2011);
Sjöholm et al. (2014) includes additional details on the short-range WindScanner system.

Based on the characteristics of the two multi-lidar instruments, the long- and short- range WindScanner systems are com-
plementary to each other. Combining these two systems into a hybrid system unlocks a possibility to simultaneously observe
mean flow features over a large region and turbulent characteristics and fine flow structures within a preselected area (not larger
than a rotor swept area) of this same region. At the current state of the lidar technology, we cannot have all these capabilities
within one single system.

The campaign Perdigão-2015, described in the present paper, was the first attempt to simultaneously operate both long- and
short- range WindScanner systems with an aim of combining them into a hybrid WindScanner system and acquiring both mean
and turbulent flow features of the site.

## 3   Methodology

The methodology for WindScanner-based experiments consists of 10 steps: (1) definition of scientific objectives; (2) site se-
lection; (3) site characterization; (4) experiment layout design; (5) scanning modes design; (6) infrastructure planning; (7)

deployment and calibration; (8) execution and data collection; (9) decommissioning and post-calibration; and (10) data archiving and dissemination.

Defining scientific objectives is related to outlining scientific questions of interest that can be addressed with WindScanners' observations. According to the scientific objectives, the site selection is made, followed by a detailed site characterization (e.g., wind conditions, terrain). Sometimes the order is reversed, a known site can stimulate a scientific question.

In the following steps, the experiment layout is made and physical infrastructure planned (e.g., power, network, access roads). Afterwards, given the now established logistical constraints, scanning modes to be implemented during the campaigns are designed. Once the physical installation commences, the deployment and calibration procedural steps are applied (e.g. leveling and orientation of WindScanner, assessment of pointing accuracy). Following the start of the campaign the execution and data collection procedural steps are put in action (e.g., experiment monitoring and information logging). The decommissioning and post-calibration procedural steps are applied at the end of the campaign. In the last procedural step, all the information regarding the campaign are collected together with the acquired datasets, and uploaded to an on-line information system, making them available for end-users.

The aforementioned steps will be demonstrated in the content of Perdigão-2015 experiment in the following section.

## 4 Perdigão 2015 implementation of methodology

There were multiple reasons in favor of Perdigão-2015 experiment: there was the need to test both the equipment and our human resources in a demanding field experiment. The question was whether the new scientific equipment, which is expensive, fragile and sensitive, and developed and tested previously in a laboratory environment or in short-duration field campaigns was robust enough to stand realistic conditions; for instance, high temperatures and remote locations with no power or network grid. The equipment, traveled by road, back and forth, between Roskilde (Denmark) and Serra do Perdigão (Portugal) and stayed in the mountains without surveillance for long periods.

WindScanner.eu (2012–2015), an European Union project of the ESFRI (European Strategy Forum on Research Infrastructures) program, under which a 3-year preparatory phase was financed for establishing a pan-European research infrastructure. Among the project deliverables, the methodology for WindScanner-based field experiments (which we report in the present manuscript) was developed based on the previous extensive work done under the WindScanner.dk project (see Vasiljević et al., 2016a; Sjöholm et al., 2014). The methodology was brought into test and further improved in a campaign held in Kassel (Germany) during the summer of 2014 (Pauscher et al., 2016; Vasiljević et al., 2016a).

Perdigão-2015 was a last demonstration campaign within the WindScanner.eu project that served as the preparation for the larger experiment conducted within the NEWA project (New European Wind Atlas, Mann et al., 2017) at the same site in 2017. Perdigão-2015 took place during the summer of 2015, from May 4 until June 29.

## 4.1 Step 1: definition of scientific objectives

Besides the WindScanner.eu project, Perdigão-2015 was funded by NEWA, Unified Turbine Testing (UniTTe), and Optimizing layouts for wind farms in complex terrain (FarmOpt) projects. The latter three projects have specific research goals that assisted in defining scientific objectives for Perdigão-2015.

The NEWA project aims to improve wind resource modeling for different site conditions. Areas with steep ridges and forested terrain are of particular interests since the current engineering (linear) flow models are unable to correctly predict the behavior of the flow over the sites with these features (Palma et al., 2008). The UniTTe project is focused on improving international standards for characterizing the wind power and load measurements of wind turbines on various sites (e.g., flat and complex terrain) by substituting mast-based measurements with measurements derived by nacelle-mounted lidars measuring close to the turbine rotor. To fulfill this project's ambition, it is necessary to investigate how the incoming flow is modified by a wind turbine. The objective of the FarmOpt project is to develop numerical tools for wind farm layout optimization in complex terrain. Here it is necessary to adequately model the wind turbine wake in complex terrain. To address the aforementioned scientific questions all three projects, thus NEWA, UniTTe, and FarmOpt call for wind field measurements in complex terrain, preferably measurements of both mean and turbulent characteristics of the flow field, where specifically UniTTe and FarmOpt require that those measurements should be done in the vicinity of a wind turbine rotor. Perdigão-2015 offered the opportunity to acquire such measurements.

For Perdigão-2015, we selected several flow aspects to investigate and addressed them with WindScanner measurements. To assist the NEWA project, we chose to measure wind resources along a ridge, occurrences of flow separations on lee sides of hills (i.e. recirculation zone) and valley flows. For the FarmOpt project, we intended to characterize a single wind turbine wake in horizontal and vertical planes. Specifically, we aimed to provide measurements for studying the wind speed deficit up to five diameters downstream of the wind turbine, the wake position in a vertical plane close to the wind turbine, and the wake geometry in a horizontal plane with center at the wind turbine hub. Similarly, for the UniTTe project, the inflow conditions were intended to be characterized in the same planes. The objective was to derive datasets for a detailed investigation of the wind turbine induction zone in complex terrain. The ultimate objective was to create a dataset that could be used in the appraisal and development of computational models for wind resource, wind turbine design and wind farm layout optimization in complex terrain.

## 4.2 Step 2: site selection

According to the scientific objectives, the site of interest should be in a terrain consisting of a hill with a steep ridge where an isolated wind turbine is operating. The site complexity should be within manageable levels, i.e. yielding a flow complexity that we can still understand. The hill size should be such that the flow clearly separates. Ideally, the hill should be in an environment surrounded by flat terrain, on which a well-defined flow would impinge, providing clear boundary conditions. Providing that this requirement is almost never met in nature, realistically, the surrounding terrain's complexity should be significantly lower than the selected site's complexity. A quasi two-dimensional, or a long ridge hill is a logical choice, and to assure a two-

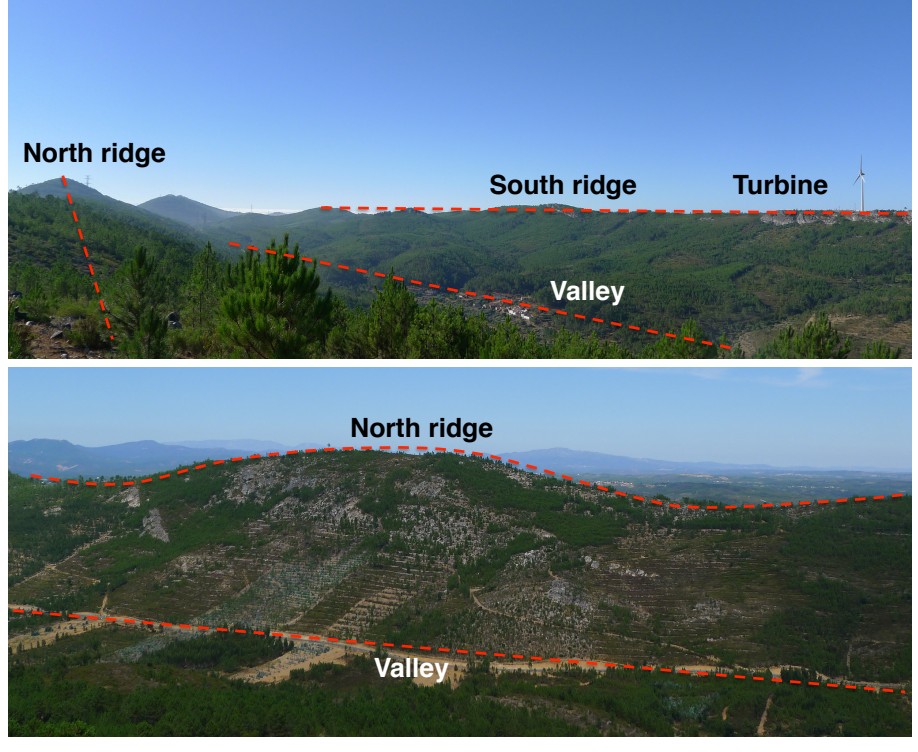

**Figure 2.** Perdigão site in September 2014 (see Vasiljević, 2014a). Views from the North ridge (top image) and South ridge (bottom image).

dimensional flow, dominant winds should be perpendicular to the ridge. Land cover, particularly forests would add to the flow complexity, and is considered desirable since many wind farms are installed near or within forested regions. According to this criteria, the Perdigão site was selected. The presence of a wind turbine at the site gave the opportunity for wake and inflow measurements.

It should be noted that the site selection is typically made considering specific criteria or the site itself triggers ideas for experiments. In case of the Perdigão site, it was both. In 2009, the site visit initiated the idea for a double-hill experiment. In 2014, due to the presence of the wind turbine, the site became an eligible location for a measurement campaign addressing the wake and inflow conditions of wind turbines in complex terrain.

### 4.3   Step 3: site characterization

Perdigão (Figure 2) is formed by two parallel ridges, with Southeast-Northwest orientation and distanced circa 1.4 km, that measure about 4 km lengthwise and are 500-550 m above sea level (asl) at their summits. The valley-to-peak height is about 200 to 250 m and the hills are steep, with an inclination of approximately 35 %. A double ridge in comparison to a single ridge site provides a list of additional advantages because the lee side flow is also the flow impinging on the second ridge. Furthermore, the lee side of the first hill and the upwind side of the second hill, make up the valley flow. The terrain coverage

is irregular, made of no or low-height vegetation and patches of eucalyptus and pine trees. Southwest and Northeast from the ridges the terrain flattens somewhat providing an environment that to a first approximation can provide definable inflow conditions. An Enercon E-82 2 MW operating wind turbine is located on the South ridge (Figure 2).

Perdigão is an ideal site in terms of the orography and main road access, but with a difficult access to the ridges. The access to ridges is mostly through low-maintained, steep and narrow unpaved roads, which creates a difficulty for the equipment

installation. During late springs and summers, the daytime temperatures are frequently above 30°C which imposes a challenge for field work. Overall, Perdigão provided a unique and demanding environment.

The site orography and canopy were mapped in March 2015 during a helicopter laser mapping mission. The area of 20 km$^2$ was scanned showing a density of about 40 points per square meter. Orthophotos with a resolution of 5 and 20 cm of the same area were acquired along with the derived point cloud. Wind measurements from the site were available from a 40-m

met mast (INEGI, 2005) from Jan 2002–Dec 2004 (3 years, Table 2), with 88% availability at location 33997 Easting, 3529 Northing and an altitude of 489 m asl at 3 heights above ground level (agl). The predominant winds were from Northeast and West/Southwest directions (Figure 3), i.e. perpendicular to the ridges, with a mean and a maximum wind speed of around 6 m/s and 20 m/s (Table 2 and Figure 3). These directions are also those with highest average velocity (Figure 3) and lowest levels of turbulent intensity.

**Table 2.** Wind characteristics (Jan 2002 – Dec 2004)

| Height (a.g.l.) [m] | 10 | 20 | 40 |
|---|---|---|---|
| Measured days | 1096 | 1096 | 1096 |
| *– Wind speed –* | | | |
| Mean (m/s) | 5.0 | 5.0 | 5.8 |
| Max (m/s) | 24.8 | 23.8 | 22.8 |
| *– Turbulence intensity –* | | | |
| $V > 5$ m/s (%) | 10.0 | 8.9 | 9.1 |
| $14 < V < 15$ m/s (%) | 8.5 | 7.7 | 8.6 |
| *– Weibull parameters –* | | | |
| $A$ (m/s) | | 5.6 | 6.3 | 6.5 |
| $k$ | | 1.8 | 2.0 | 2.0 |

**4.4   Step 4: experiment layout design**

Computer simulations (Gomes, 2011) based on the VENTOS® (Castro et al., 2003) code of the flow over Perdigão were of importance while designing the experiment layout. VENTOS® is a dedicated CFD code for solving ABL flows over complex terrain, based on the elliptical RANS equations and the k-$\varepsilon$ model for turbulence closure. The flow domain is described with a terrain-following structured mesh, with the bottom boundary modeled as a rough surface wall, classic inlet and outlet boundary conditions, symmetry condition imposed at the lateral boundaries and a zero-shear condition for the top boundary. The terrain

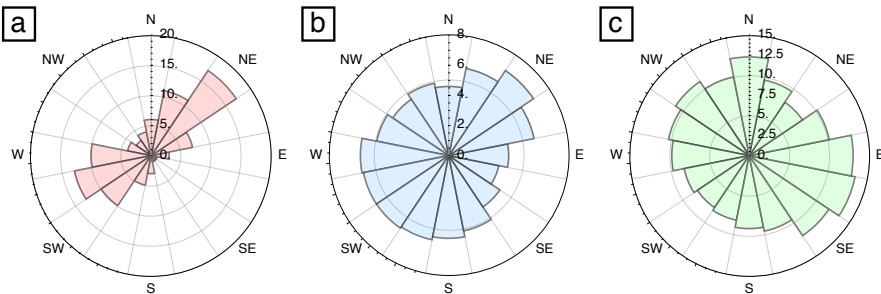

**Figure 3.** Wind regime at met mast station for 3-year period at 40 m agl: (**a**) wind direction (%); (**b**) wind speed (m/s); (**c**) turbulence intensity (%)

map of Perdigão was sourced from SRTM data (90 m×90 m local horizontal resolution) combined with 10m resolution height contour chart data closer to the site. Domain meshes covering a 20 km×20 km area around the site were produced, discretized using a grid of 100×100 nodes measuring a minimum of 20 m×20 m at the meteorological mast position of previous wind resource studies (Table 3). Domain ceiling was set at 3000 m asl, discretized with 60 nodes, the smallest at 2 m high at the ground surface. No canopy areas were considered. The solution was obtained using steady state formulation for the equations, and the model's wall boundary parameters were calibrated to yield wind speeds of 5-6 m/s at the South ridge under Southwest winds, with a surface characteristic roughness $z_0$ set to 0.03, friction velocity $u_*$ to 0.23 m/s. The simulations of the flow from the Northeast and Southwest predict a high complexity of the flow (Figure 4), and large recirculation zone enclosed in the valley (Figure 5).

The two WindScanner systems, comprised a total of six scanning lidars: three LRWSs named Koshava (LR1), Sterenn (LR2) and Whittle (LR3), and three SRWSs, named R2D1 (SR1), R2D2 (SR2) and R2D3 (SR3). The lidar locations were selected according to the aim to sample the flow field along the South ridge and within the transect perpendicular to the ridges that goes through the wind turbine (Figure 6 and Table 3), while taking into account an intersecting angles between laser beams and elevation angles at which the laser beams will be directed.

We define the intersecting angle as the smallest angle between the projections of two intersecting laser beams in a horizontal plane. The intersecting angle can take any value between $0°$ and $90°$. When selecting lidar locations we intend to have an intersecting angle of at least 30 degrees respect to the prevailing wind direction. Based on a simple accuracy model (see Vasiljević and Courtney, 2017) the intersecting angle of $30°$ results in the accuracy of about 0.25 m/s for the retrieved horizontal wind speed. In order to accurately retrieve the vertical wind speed the elevation angles should be as steep as possible, preferably $90°$ as indicated in (Debnath et al., 2017). Hence, based on this study results, which uses a norm approach described in (Simley et al., 2016) to assess the suitability of the multi-Doppler setup, the elevation angles larger than $45°$ provide means to accurately acquire the vertical component.

The previously acquired point cloud and orthophotos assisted in choosing the most accessible locations for the WindScanners' installation respect to the previously established criteria. LR3 was located on the South ridge next to the wind turbine,

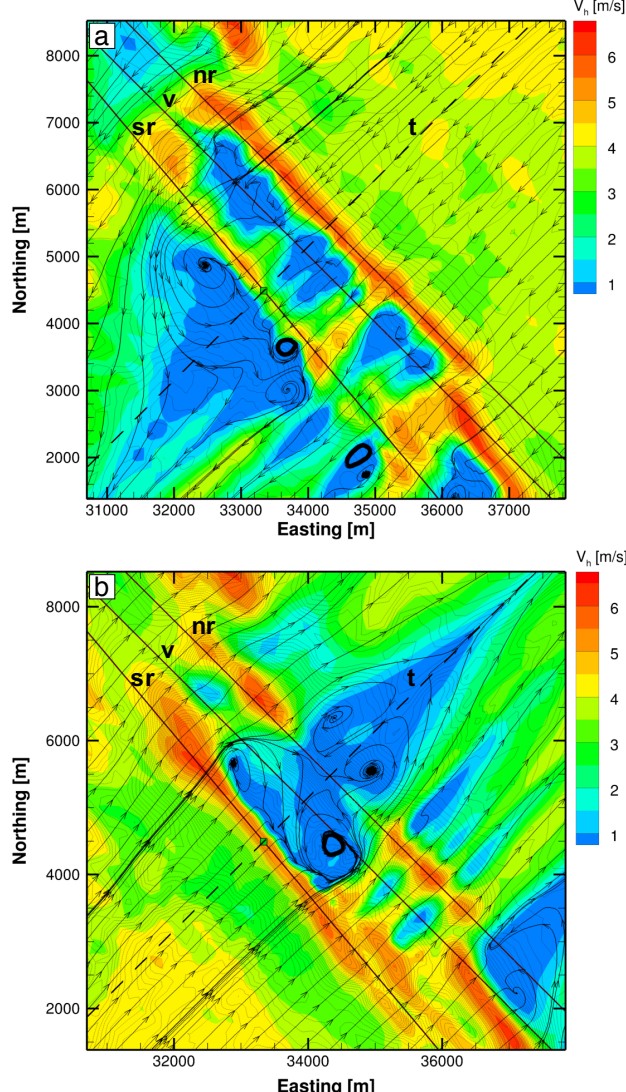

**Figure 4.** Simulated wind flow on a surface 80 m agl: (**a**) Northeast winds; (**b**) Southwest winds. Positions expressed in datum ETR89/PTM06 (m) (EUREF, 2016). Three thick diagonal lines denoted as *sr, v and nr* represent South ridge, valley and North ride lines. Dashed line denoted *t* represents transect of interest.

and LR1 and LR2 were located on the North ridge (Figure 6a). The distance from LR3 to LR1 and from LR3 to LR2 was about 1.5 km, while the distance between LR1 and LR2 was about 1.2 km. The three SRWSs were located on the South ridge close to the wind turbine (Figure 6b). SR1 and SR3 close to the access road, 52 m Southeast and 45 m Northwest of the wind turbine respectively, and SR2 43 m Northeast of the wind turbine, down the slope of the South ridge (Table 3 and Figure 6b).

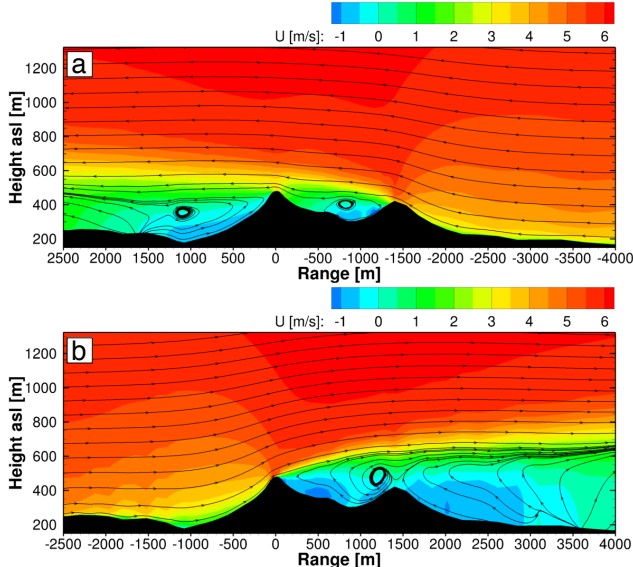

**Figure 5.** Simulated wind flow in a vertical plane indicated with the dashed line in Figure 4: (**a**) Northeast winds; (**b**) Southwest winds. Coordinate system origin corresponds to wind turbine base.

**Table 3.** Position of instruments and landmarks given in datum ETR89/PTM06 (m)

| Landmark / Instrument | Easting [m] | Northing [m] | Height [m] | Distance to turbine [m] | Direction to turbine [°] |
|---|---|---|---|---|---|
| Koshava – LR1 | 34793.95 | 4806.36 | 450.15 | 1553.37 | 76.30 |
| Sterenn – LR2 | 33983.37 | 5722.43 | 443.71 | 1462.11 | 28.54 |
| Whittle – LR3 | 33298.12 | 4430.08 | 485.58 | 15.50 | 122.82 |
| R2D1 – SR1 | 33282.19 | 4491.10 | 486.19 | 52.79 | 356.77 |
| R2D2 – SR2 | 33327.27 | 4437.05 | 474.83 | 43.12 | 91.89 |
| R2D3 – SR3 | 33285.10 | 4393.12 | 487.27 | 45.43 | 180.08 |
| Wind turbine | 33285.16 | 4438.44 | 484.01 | – | – |
| Calibration pole | 33314.46 | 4433.14 | 487.01 | 29.92 | 100.253 |
| Telecom tower | 33287.29 | 4411.11 | 502.70 | 33.19 | 175.54 |
| Power tower | 34791.18 | 4820.50 | 512.30 | 1553.98 | 75.77 |

## 4.5   Step 5: infrastructure planning

The field campaign's control center, consisting of the LRWS and SRWS master computers, data processing computer, network appliances, WiFi antennas, security camera and a small work space was installed in an office container, 20 m Northeast from the wind turbine. Power and Internet connections were provided from the wind turbine substation by pulling 200 m long power

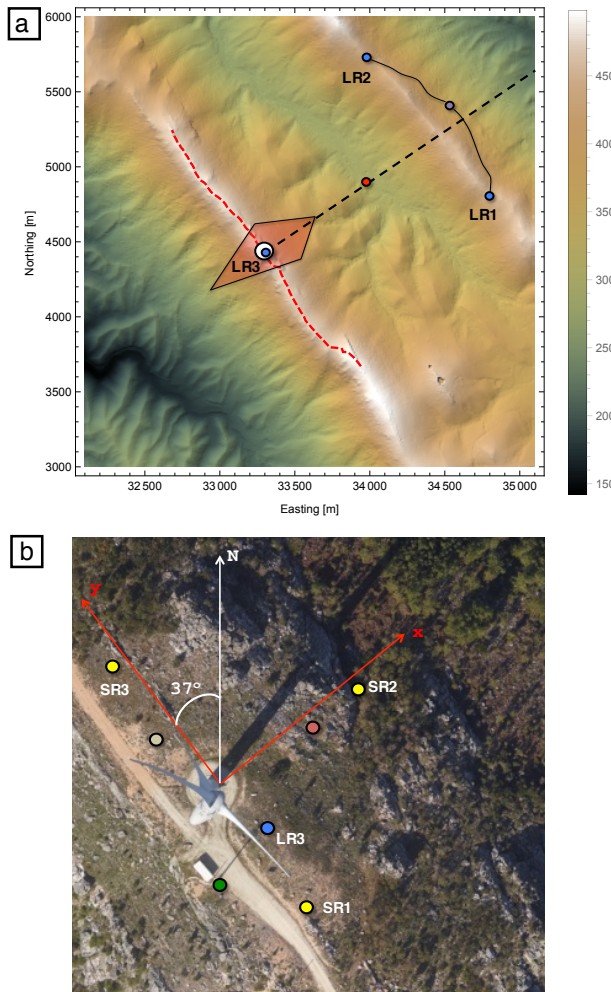

**Figure 6.** Perdigão site: (**a**) Overall view (blue circles, long-range WindScanner units; white circle, wind turbine; pink quadrilateral, diamond scan; dashed black line, transect scan; red circle, virtual mast scan; solid black lines, power cables; red dashed line, ridge scan); (**b**) Top view around the wind turbine (yellow circles, short-range WindScanner units; blue circle, long-range WindScanner unit LR3; grey circle, control centre; red circle, calibration pole; green circle, telecom tower and substation)

and fibre-optic cables from the substation to the control center, from which the wired power and Internet connections were further distributed to the nearby WindScanners. For network redundancy, secondary Internet connection for all devices was provided via mobile network connections. For the uninterrupted communication and power of LR1 and LR2 on the opposite (North) ridge, two unidirectional WiFi links from the control center were configured and a power container located on the Northeast slope of the North ridge approximately 1 km from each lidar unit (Figure 6). The power container included two diesel generators and a fuel tank large enough for a one-week operation without refueling. At any moment, only one diesel generator was operational, while the second one served as a fallback solution. The WindScanner locations were prepared by the

local municipality to ensure a well-leveled base for the deployment. It should be noted that without the local support achieving the aforementioned experiment layout would be almost impossible (see Vasiljević, 2015a, for more details) .

### 4.6   Step 6: deployment and calibration procedures

After all scanning lidars were positioned, oriented, and leveled at the designated locations (see Vasiljević, 2015b), their absolute positions were acquired using a multi-station (combination of total station and GPS) in the conjunction with DGPS/RTK correction data (accuracy about 1 cm). The correction data were provided by the Geographic Institute of the Portuguese Army. Additionally, the absolute position of several landmarks (e.g. wind turbine, telecom tower) were acquired and used for the
WindScanner' calibration.

For assessing the pointing accuracy of LRWSs, selected landmarks were mapped with the LRWSs laser beams using the CNR (carrier-to-noise) mapper (pg. 157, Vasiljević, 2014b). The power line tower on the North ridge was mapped with LR3, whereas the telecom tower next to the wind turbine was mapped with LR1 and LR2. The comparison of the referenced and mapped positions indicated a pointing accuracy of $0.05^{\circ}$ in azimuth and elevation for all three LRWSs. From the CNR maps, we
were able to determine the backlash level of $0.025^{\circ}$, which is consistent with results from previous deployments (see Vasiljević et al., 2016a). The sensing range accuracy was assessed using the same landmarks. The laser beams were steered to hit the landmarks, and we assessed the range gate positions (i.e., distances along LOS) at which the intensity of the backscattered light (CNR) was maximum since these positions correspond to the distances between the landmarks and LRWSs. We found sensing range offsets of 8, 10 and 3 m for LR1, LR2 and LR3 respectively, which were corrected for during the design of the scanning
methods.

Since the DC component in the SRWS Doppler spectra is notched out, non-moving hard targets cannot be used to assess their pointing and sensing range accuracy. Instead of non-moving targets, for the SRWSs we employed a 12 m calibration pole with two motor-driven balls on the top made of low-density polyether (Figure 7a). The rotating balls were scanned simultaneously from the three SRWSs by mapping three separate 2 m×2 m vertical planes, oriented perpendicular to the pointing direction
of each instrument. In Figure 7b, the result of such a scan is presented, where only the signals from the two rotating balls are highlighted. When the mapped positions of the balls were compared with the reference positions, we found similar pointing accuracy level of the SRWSs to that of the LRWSs. The rotating balls were also used to assess the sensing range accuracy. The three laser beams were steered to meet the center of the top rotating ball and the focus points were moved along the laser beam propagation paths. The results of this test, Figure 8, depict the intensity of the backscattered signal of each of the SRWSs versus the distance of focusing, and the theoretical distribution of the intensity based on the assumption that the intensity of a focused laser beam follows a Lorentzian distribution. It is assumed that the maximum intensity appears at the distance where the moving target is located in regards to a SRWS. It has been found that the calculated and measured distances corresponded
to each other to within a few centimeters.

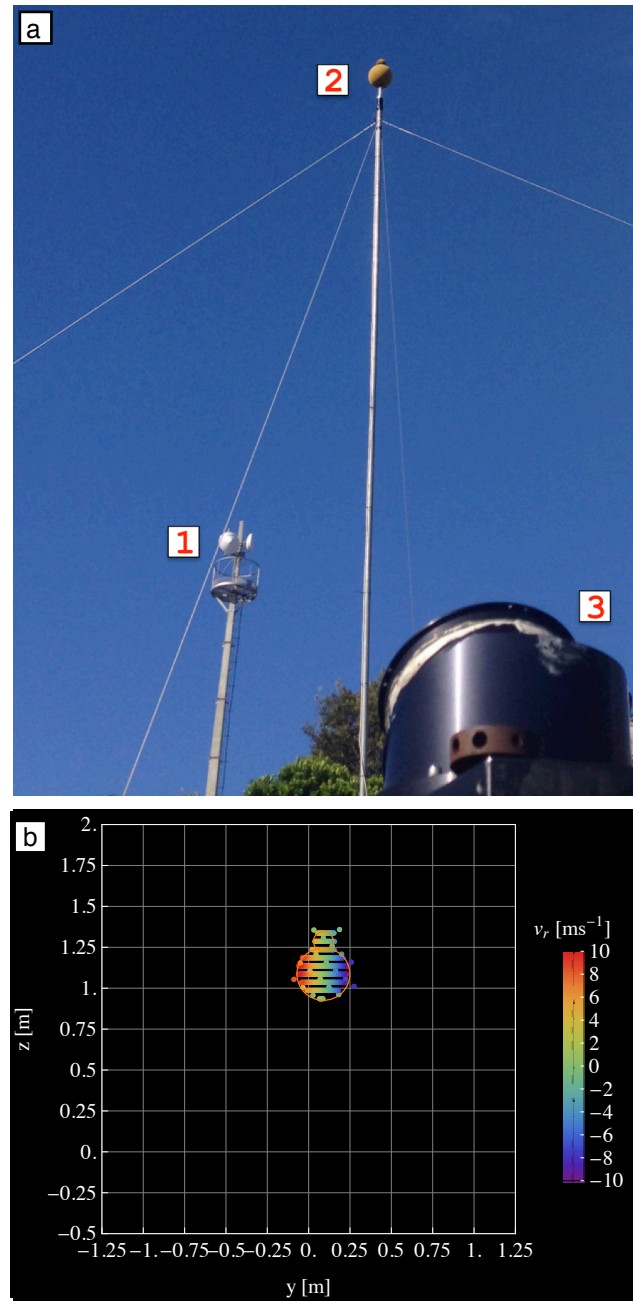

**Figure 7.** Calibration of SRWS: (**a**) actual setup (1 - telecom tower, 2 - calibration pole with two motor-driven balls and 3 - SRWS's scanner head); (**b**) radial wind speeds induced from the rotation of the two rotating balls as measured by SR1 (the perimeter of the two balls is depicted with orange circles)

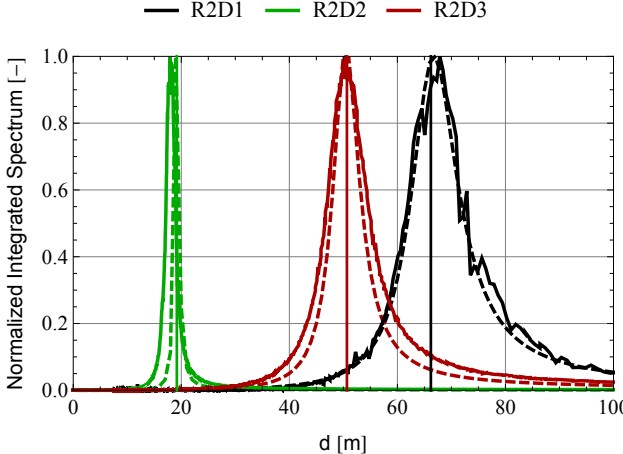

**Figure 8.** Integrated spectrum versus focus distances for SR1 (black line), SR2 (green line), and SR3 (red line)

### 4.7 Step 7: scanning modes design

Five scanning modes, Table 4, were designed to investigate the flow details around the wind turbine and in the valley: ridge, diamond, and transect scans in the case of the LRWS, and T and vertical plane scans in the case of the SRWS.

The ridge scan was designed to address the flow above the South ridge (Table 4 and Figure 6). The two LRWSs, LR1 and
LR2, were configured to intersect their laser beams and to move the beam intersection along a curved line 2 km long, 80 m above the crest (i.e. the wind turbine hub height), following the terrain profile (see Vasiljević, 2016c). The line was designed using the point cloud information. The mean elevation angle at which both laser beams were directed was 4.11°. Throughout the scan, on average the intersecting angle between the two laser beams was 42.21°.

To investigate the inflow and wake conditions of the wind turbine at a large scale, LR1 and LR2 were configured to guide
the laser beams within a 4.7° vertically inclined quadrilateral with the center at the turbine hub, and dimensions of 500 m × 750 m. The quadrilateral lies on an inclined horizontal plane defined by LR1, LR2 and hub position. The wind turbine hub represented the middle point of a 1-km long diagonal of the quadrilateral. Along this diagonal, LR1 and LR2 synchronously intersected laser beams and move the beam intersection resolving instantaneous horizontal wind speed and wind direction (true dual-Doppler) at 50 points, thus every 20 m (see Vasiljević, 2016a). Since the beams were guided within the same plane, this
allowed retrieval of horizontal wind speed and wind direction (unsynchronized) in an additional 2450 measurement points (Table 4 and Figure 6). In these additional points, LR1 and LR2 acquire radial wind speed asynchronously. The scan mode was named the diamond scan. It is important to notice that the diamond scan does not consist of two synchronized plan position indicator (PPI) scans, but of two synchronized user-defined scans. Moving a laser beam within an inclined horizontal plane requires variable elevation angle which is a function of the azimuth angle. Since the wind characteristics are such that the wind direction is mainly either Northeast or Southwest, the diamond scan can be used to measure simultaneously the inflow
and wake. The objective with the diamond scan was to obtain data for studying the wind speed deficit up to five diameters

downstream of the wind turbine, the wake position and wake geometry in a horizontal plane with the centre at the wind turbine hub, as well as the inflow conditions up to five diameters upstream of the wind turbine. The average intersecting angle between the two laser beams during the diamond scan was $49.37^{\circ}$, while the average elevation angle at which the beams were directed was $4.33^{\circ}$.

In addition to the large-scale flow observations, two scanning modes for the short-range WindScanner system (Table 4 and Figure 9) were configured to provide insights on the inflow and wake at the turbine scale. The T-scan was used to measure the turbine inflow conditions. Two different versions of this mode were used, differing in the dimensions of the scanning area. Both of them consisted of one vertical and one horizontal plane that were perpendicular to each other and perpendicular to the South ridge line. In Figure 9a, we can see the first version in the Cartesian coordinate system, with the origin in the wind turbine base

rotated $37^{\circ}$ clockwise from North. The area covered with the first scan mode version was from -56 m to 56 m along the y-axis, 0 m to 80 m along the x-axis, and from 14 m to 78 m along the z-axis. Within the enclosed area, 15 horizontal and 9 vertical line scans were performed. The second version covered an area between $-64$ m to 64 m along the y-axis, $-10$ m to 100 m along the x-axis, and 14 m to 78 m along the z-axis, encompassing 17 horizontal and 9 vertical line scans. On average, the intersecting angles between the SR1 and SR2, SR1 and SR3, and SR2 and SR3 laser beams were $44.26^{\circ}$, $55.78^{\circ}$ and $49.64^{\circ}$

respectively. The average elevation angle at which the beams were directed was $50.92^{\circ}$.

    To address the wake conditions with the short-range WindScanner system the vertical scan mode consisting of vertical lines was employed, Figure 9b. Also, two different versions of the scanning mode were used. Both of them focused on scanning in vertical lines placed in a plane parallel to the wind turbine's rotor. The scanning plane was confined in the y-z plane, between $-48$ and 48 m along y, and between 30 and 128 m along z. In the first version of the scanning pattern, the vertically scanned

lines were located in a vertical plane one rotor diameter (80 m) away. In the second version, one more plane with the same dimensions was added half a rotor diameter in the downwind direction. On average, the intersecting angles between the SR1 and SR2, SR1 and SR3, and SR2 and SR3 laser beams were $28.77^{\circ}$, $59.96^{\circ}$ and $49.21^{\circ}$ respectively. The average elevation angle at which the beams were directed was $49.91^{\circ}$.

    While executing the second version of the T and vertical scan modes we observed a lag (loss of synchronization) between

SR3 respect to the other two SRWSs (SR2 and SR1), which operated in sync. The lag linearly increased over the time with a constant rate of 16.67 $\mu$s / s and 5000 $\mu$s / s for the T and vertical scan modes respectively. The observed lag was caused by the safety limits of the scanner head controller, which didn't allow higher scanning speeds than the maximum permitted speed for a safe operation. As the result, SR3 was lagging behind SR2 and SR1. The reset of the scanning modes zeroed the lag, which then once again increased over time as described. The reset took place each time when we switched from one scanning mode to

another. However, this was not done in a systematic way since the scanning mode switching took place when the wind direction changed from the Northeast to Southwest. Therefore, the values of the maximum lag for the short-range WindScanner system in Table 4 are only given for the first version of the T and vertical scan modes since in the first version of scans the SRWSs were in sync.

    The transect scan was employed to investigate the flow field within a vertical plane perpendicular to the ridges entailing

5  the wind turbine (Table 4, Figures 6 and 10). The three LRWSs were configured to perform intersecting and synchronized

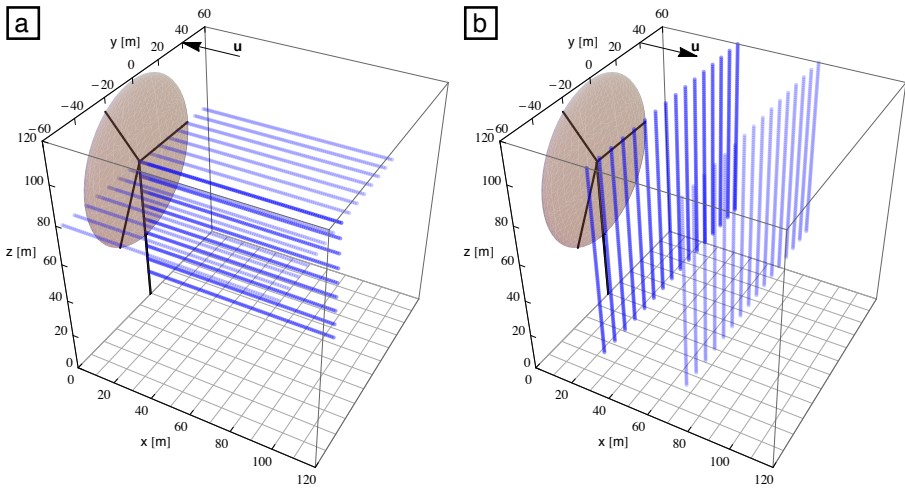

**Figure 9.** Scanning patterns of short-range WindScanners: (**a**) T-scan; (**b**) vertical planes.

Range-Height Indicator (RHI) scans. The RHI scan of LR3 coincided with the vertical plane (Figure 10), whereas other two RHI scans sliced through this plane along a 500-m vertical line position in the valley (see the vertical dashed line in Figure 10). The radial wind field measured by LR3 provided the possibility to assess the recirculation zone on lee sides of either ridge in the case of Southwest and Northeast winds, traces of the wind turbine wake, and the flow in the valley (see Vasiljević, 2016d). The intersection of the three RHI scans represented a 500-m virtual mast. The intersecting angle during the scan between the LR1 and LR2 laser beams was $84.10^{\circ}$, while this angle between the LR2 and LR3 laser beams and between the LR1 and LR3 was equal to $54.77^{\circ}$ and $41.12^{\circ}$ respectively. The elevation angle at which the beams were directed was in a range from about $-12^{\circ}$ to $23^{\circ}$. Even though we had three independent LOS measurement along the virtual mast due to low elevation angles only
the horizontal components of the wind vector (i.e., $u$ and $v$) were considered in the data analysis (see explanations in Berg et al., 2015; Debnath et al., 2017).

    The long-range WindScanner system's scans were run in a batch mode, where each strategy was executed over a 10-min period. The sequence of the ridge, diamond and transect scan was executed over a 30-min period and then repeated. Because only LR1 and LR2 were needed to execute the ridge and diamond scans, LR3 continued to perform the transect strategy
throughout the entire campaign.

    The Portuguese weather service (IPMA) provided a daily forecast for the Perdigão site using a non-hydrostatic numerical weather prediction model of a limited area (AROME, the numerical prediction model of Meteo-France) for variables such as temperature, relative humidity, rain, wind direction and velocity at 10 m and 80 m agl. The wind direction was used to decide the scanning mode of the short-range WindScanner system: the vertical mode if the wind was coming from Southwest direction
(wake scanning) or the T-scan mode in the case of Northeast wind (inflow scanning).

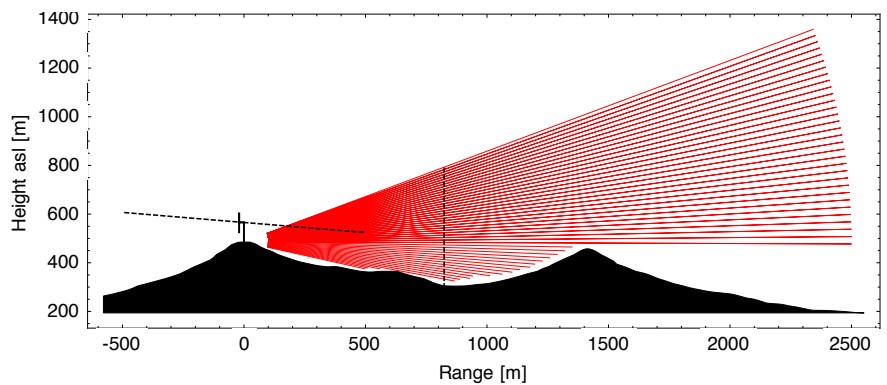

**Figure 10.** Transect Scan: red lines indicate LOS measurements acquired by LR3, vertical dashed line shows the position of the virtual mast, and inclined horizontal dashed line the position of the diamond scan plane

**Table 4.** Scanning modes

|  | LRWS system | | | SRWS system | |
|---|---|---|---|---|---|
| Scanning mode | Ridge | Diamond | Transect | T | Vertical |
| WindScanners | LR1 | LR1 | LR1 | SR1 | SR1 |
|  | LR2 | LR2 | LR2 | SR2 | SR2 |
|  |  |  | LR3 | SR3 | SR3 |
| Variables | $u, v$ | $u, v$ | $u, v, v_{LOS}$ | $u, v, w$ | $u, v, w$ |
| LOS sampling (Hz) | 2 | 2 | 2 | 100 | 100 |
| Points per scan | 100 | 2500 | 50 $u,v$, 12000 $v_{LOS}$ | 350 | 312 |
| Scans per 10 min | 12 | 24 | 24 | 9 | 9 |
| Probe length (m) | 35 | 35 | 35 | 0.7–38 | 1.7–47 |
| Synchronization (Max lag (ms)) | 10 | 10 | 10 | 20 | 20 |
| Dynamic ranging | Yes | Yes | Yes | – | – |
| Dynamic focusing | – | – | – | Yes | Yes |

## 4.8 Step 8: execution and data collection

Throughout the course of the experiment visits to the WindScanners and hardware checks were performed several times per day in the first three weeks of the experiment, as fine adjustments of the scanning modes were needed, and then every other day in the last stage of the experiment. Also, the staff at the site during the first three weeks decreased from ten to one single person, who managed the whole setup until the end. The status of the experiment was reported daily on the dedicated web blog (see Vasiljević, 2015a, for more details). The data were collected every third day and transmitted to the database at DTU. Preliminary data analysis was done on the fly for quality and quantity assessing. This assessment was of particular importance

since it helped in making a decision in extending the length of the experiment from the originally planned four weeks to the total of eight weeks.

Although, this was the first time the long- and short- range WindScanner systems were used simultaneously, because of several reasons we only collected a few periods of simultaneous measurements. Despite the rapid installation of the Wind-Scanner systems, the off-grid power solution for the long-range WindScanner system was deployed roughly ten days after the short-range WindScanner system started collecting data. An issue with the control unit of one of the LRWSs forced us to reject the measurements during May 23–25 (see the blog post for these dates). The cooling systems of long and short-range WindScanners could not cope with the high temperature around midday ($\geq 40^{\circ}$C) and we had to pause the measurements for a couple of hours every day. We discussed this issue more in (Vasiljević et al., 2016a) where we suggested possible solutions. The short-range WindScanner system had a technical issue with the MACRO ring which halted the short-range WindScanner observations during June.

## 4.9    Step 9: decommissioning and post-calibration procedures

At the end of the measurement period the pointing and sensing range accuracy were reassessed with the LRWSs only. We found that the new positions of the landmarks matched the previously mapped position within $0.05^{\circ}$ in azimuth and elevation for all three LRWSs. The backlash levels remained the same, $0.025^{\circ}$. The new set of sensed distances showed no difference comparing to the previously acquired set. Due to technical issues of the SRWSs, the post-calibration procedure was not performed for this system. After the post-calibration procedure the WindScanners were deinstalled, and all traces of their presence removed from the sites.

## 4.10    Step 10: data dissemination and availability

The short-range WindScanner system was operational from May 8 until June 3. During this period, 110 hours (665 runs with a duration of 10-minute each) of data were acquired. A total of 407 runs were made with the T-scan mode, addressing inflow conditions, whereas the remaining 258 were made by employing the vertical planes mode, addressing wake conditions. The long-range WindScanner system was operational from the May 19 until June 26. Overall, the long-range WindScanner system acquired 528 hours of data, of which about 180 hours of data were recorded with each scanning strategy. A special case is the transect scan, where besides 180 hours of concurring data acquired with all three LRWS, there is also the additional 360 hours of data collected only by LR3. There are 6 hours of simultaneous measurements of the inflow conditions with both WindScanner systems.

Also, for the entire measurement period (May-June) the owner of the wind turbine (Generg) provided 10-minute means of the wind turbine SCADA data.

The acquired datasets of radial velocities were entirely processed and data artifacts removed. The LRWS dataset was filtered on the CNR values of each individual measurement point ($CNR > -27$ dB). Besides the filtering, a multi-range gate analysis was applied to remove remaining spurious data points where, using multi-peak detection in the CNR values for each LOS, data points contaminated with hard targets were removed. Also, by detecting discrete jumps in the radial speed values along

**Table 5.** Data availability expressed in hours

| | LRWS system | | |
|---|---|---|---|
| Scanning mode | Ridge | Diamond | Transect |
| Data before filtering on sampling | 187 | 185 | 181 |
| Data after filtering on sampling | 77 | 94 | 116 |
| of which in Northeast sector | 7.5 | 10 | 15 |
| of which in Southwest sector | 19 | 16 | 12 |

each LOS, errors in the spectral estimation were rejected from the datasets. Details about the multi-range gate analysis will be presented in a separate publication.

The SRWS data were first filtered to remove signals that originated from hard target motions (i.e. wind turbine blades). These signals are relatively easy to detect due to the corresponding high-intensity power spectral density that appears in the Doppler spectra, as a result of the increased backscattered light from hard targets, relative to the one from aerosols. Afterwards, the data are filtered by using an adaptive threshold of acceptable maximum and minimum of the total energy of the laser Doppler spectra. The threshold value is calculated by the lower and upper outer fences of the distribution of the total energy per spectrum. Consequently, the data were spatially averaged after being grouped in cubic grid cells both of $4\times4\times4$ m and $8\times8\times8$ m dimensions. To treat the synchronization issues that appeared in the second version of T-scan and Vertical plan patterns, the data acquired using those pattern versions were additionally time averaged in 10-minute periods.

Where possible, the processed radial velocities were combined to calculate two (diamond, ridge and transect scanning methods) or all three components of the wind vector (T and vertical scanning method). Details about the retrieval techniques are given in Appendix A. Afterwards, wind vector and radial fields were averaged over a 10-minute period and plotted. The resulting figures were saved as PNG files, available for users to visually browse the dataset. To indicate the amount of high-quality data collected with the long-range WindScanner system, we applied a simple filter on sampling constraints after processing all data, which consisted of selecting only those 10-minute runs where there was at least 50% of maximum number of scan iterations and at least 50% of the maximum number of measurement points. The results are given in Table 5 only for the long-range WindScanner system. Currently, for the SRWS, each 10 minute period is being manually evaluated in order to determine whether the wake or induction zone are captured with measurements. This is the reason why Table 5 does not include information on the SRWS data.

Both, raw and highly processed datasets have been uploaded to a MySQL database and are currently available for the participants of the projects that funded the experiment. We intend to release the entire dataset for public use during the second half of 2017 through the e-WindLidar web platform (http://e-windlidar.windenergy.dtu.dk). At present, the data is being analyzed by the research groups directly involved in Perdigão-2015, and has been presented in appropriate forums. Examples of those presentations and publications are for instance, Mann et al. (2016) on the details of the experiment, Rodrigues et al. (2016)

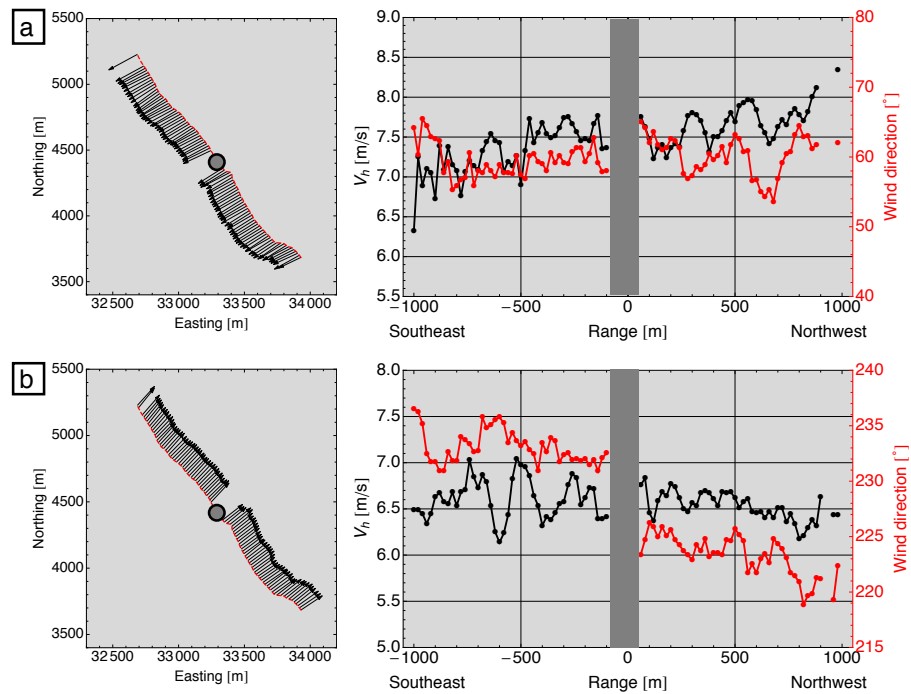

**Figure 11.** Wind vectors, wind speed and direction (10-min averaged), 80 m agl along the South ridge for wind dominant directions acquired using the ridge scan performed by the long-range WindScanner system: **(a)** Northeast (measurements taken on June 7 2015 from 03:30 to 03:40 UTC); **(b)** Southwest (measurements taken on June 21 2015 from 21:42 to 21:52 UTC). Black line, horizontal wind speed; red line, wind direction; dashed red line, ridge scan; grey circle (left) and rectangle (right) areas where measurements are erroneous due to the presence of the wind turbine.

focused on the flow in the valley and over the two-ridges, Hansen et al. (2016) on the wind turbine wake, and Meyer Forsting et al. (2016) on the inflow conditions.

# 5   Discussion

## 5.1   Observational results

Figures 11 to 15 intend to illustrate the performance of the scanning modes and also be a sample of the observations and flow phenomena being present.

The ridge scan, 2 km along the South ridge, shows no turning of the wind for Northeast winds (Figure 11a). Thus, for this wind direction we observed a two-dimensional flow. On the other hand, for Southwest winds there is a slight turning of the

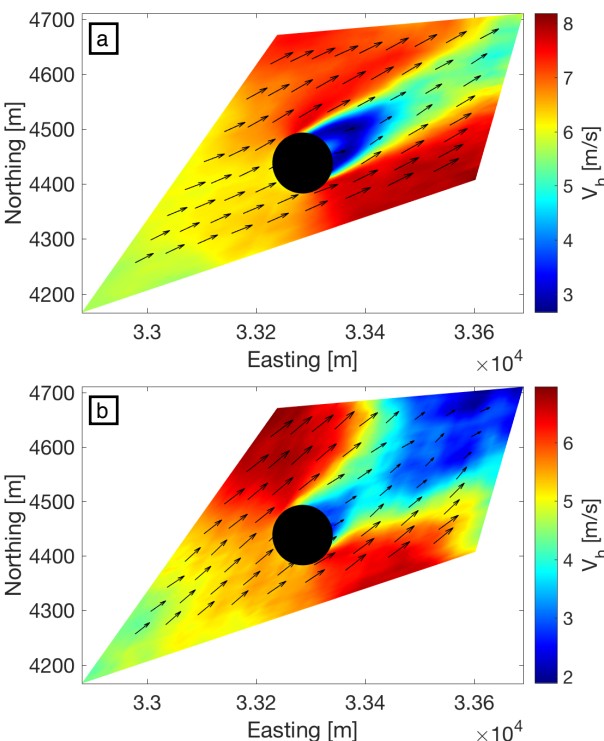

**Figure 12.** The wake in a horizontal plane observed with the diamond scan (10-min averaged) performed by the long-range WindScanner system for different atmospheric conditions: **(a)** stable conditions (measurements taken on June 10 2015 from 22:20 to 22:30 UTC); **(b)** unstable conditions (measurements taken on June 11 2015 from 16:02 to 16:12 UTC). The black circle represents rotor swept area.

wind (the difference between the wind direction at the edges of the transect is roughly $15^{\circ}$, Figure 11b). The maximum wind speed along the transect is not at the wind turbine location regardless of the dominant wind direction (Figures 11).

The initial data analysis of the wake measurements indicates a clear diurnal dependence of the wake characteristics (see Hansen et al., 2016), which may be related to the stratification. A well-formed, narrow and long wake was pulled down the slope during late nights until early mornings when we expected stable conditions and reduced mixing (Figures 12a and 13a). On the other hand, during the rest of the day under more unstable conditions and increased mixing, the wake was wider, shorter and lifted up (Figures 12b and 13b). The inflow and wake of the turbine during one full day is well represented in Vasiljević (2016b).

The objective of the transect scans (Figure 14) was the mapping of the flow over the ridges and in the valley. For the dominant wind directions Northeast and Southwest (Figures14a and 14b) the recirculation zones on the lee sides (south side of the North ridge and north side of the South ridge) of the hills can be seen, as suggested by the computational results in Figure 5. Although sharing a common feature, these two figures correspond to two different days (7 and 10 of June) and two different times (03:29 and 13:43 UTC), night and day at a time of low and high temperature. The velocity field at 03:19, Figure 14a displays a pattern,

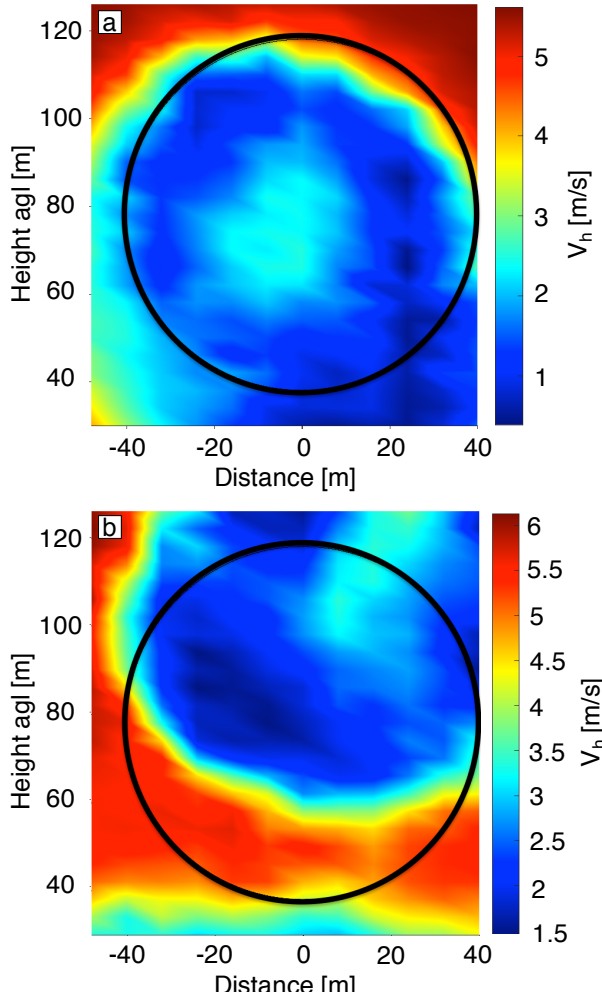

**Figure 13.** Wind speed (10-min averaged) wake in a vertical plane one rotor diameter away from the turbine observed with the vertical planes scans performed by the short-range WindScanner system: **(a)** wake pulled down (measurements taken on June 12 2015 from 00:40 to 00:50 UTC); **(b)** wake lifted up (measurements taken on June 11 2015 from 19:10 to 19:20 UTC). Black circle, rotor swept area.

a two-layer atmosphere of maximum wind speed at around 500 m and and 750 m, and a constant wind speed between 500 and 650 m. Figure 14c, around the same time in the night, displays a stratified atmosphere with a well-defined internal wave, originating in the South ridge, of a length equal to the distance between ridges (see 24H animation Vasiljević, 2016e). This phenomenon and the conditions under which it can persist, although a subject in Rodrigues et al. (2016), will be studied further.

Figure 15 illustrates hybrid WindScanner measurements of the wind turbine inflow conditions. We can see combined measurement planes acquired by the long-range WindScanner system using the diamond scan (horizontal plane) and the short-range WindScanner system using the vertical plane scan (vertical plane). During the depicted observational period the wind direction

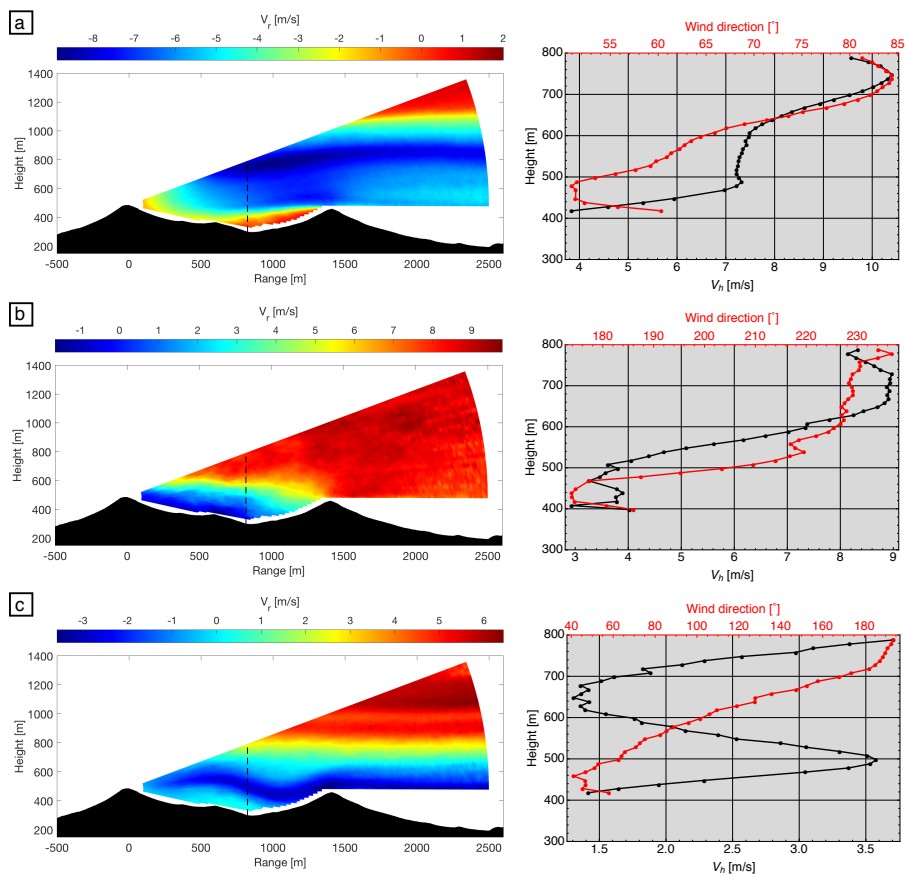

**Figure 14.** Radial flow fields (10-min averaged) in a vertical plane (left figures) and corresponding virtual mast measurements of the horizontal wind speed and wind direction (right figures) acquired using the transect scan performed by the long-range WindScanner system: **(a)** Northeast dominant wind, recirculation zone on the North ridge (measurements were taken on June 7 2015 from 03:19 to 03:29 UTC); **(b)** SouthWest dominant wind, recirculation zone on the South ridge (measurements were taken on June 10 2015 from 13:33 to 13:43 UTC); **(c)** Northeast dominant wind, atmospheric internal wave (measurements taken on June 10 2015 from 03:32 to 03:42 UTC). Negative and positive radial wind speeds indicate winds going towards and away from the lidar, black and red solid curves correspond to the horizontal wind speed and wind direction, and the dashed black line indicates a position of the virtual met mast. Note that color pattern of the radial velocity field (left) is not the same in every figure.

was approximately 45°, thus the rotor plane was not completely aligned with the vertical plane scan (53°), while the mean horizontal wind speed was about 4.5 m/s at the hub height. The comparison of the long- and short- range WindScanner system horizontal wind speed measurements at horizontal lines closest to the two scanned plane intersection shows generally a good agreement (the averaged mean difference about 0.2 m/s). We should not expect the exact match between these measurements

since how the flow was probed at the coinciding points with two WindScanner systems differs. Throughout the diamond scan the laser beams of the long-range WindScanner system were steered within a horizontal plane, whereas the laser beams of

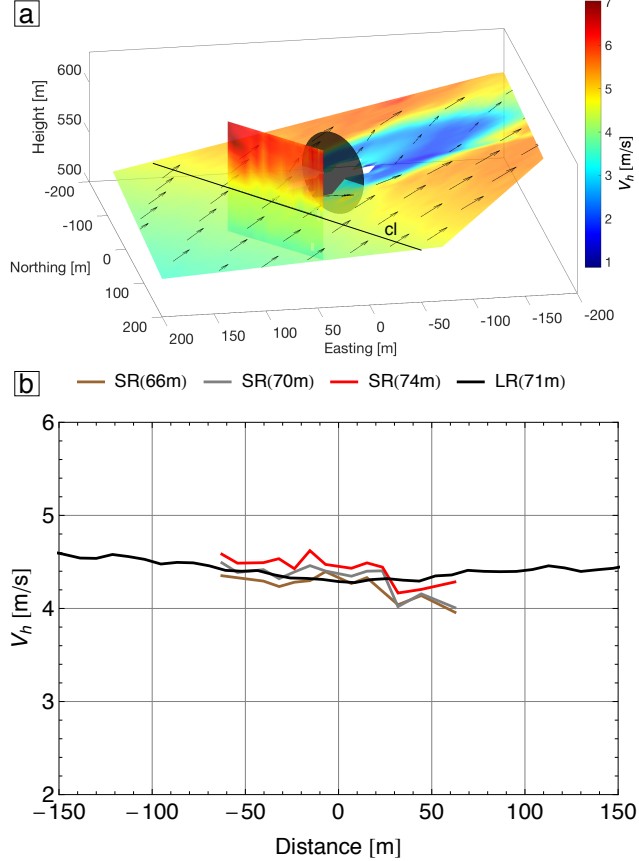

**Figure 15.** Inflow conditions (10-min averaged) measured by a hybrid WindScanner system on June 3 2016 from 07:30 to 07:40 UTC: **(a)** three-dimensional repersentation of two scanned plane (vertical plane – SRWS, horizontal plane – LRWS) with a black line denoted *cl* representing the line where the planes intersection coincides; **(b)** the comparison of horizontal wind speed measured by the short-range WindScanner system (lines denoted by *SR*) and long-range WindScanner system (line denoted by *LR*) at points distributed along several horizontal lines (lines at 66, 70, 71 and 74 m agl) which are closest to the planes intersection denoted cl in the upper figure

.

the shot-range WindScanner system during the vertical plane scan were directed with steep elevation angles (the averaged elevation angle was about 50°) . Besides the difference in the positioning of the probes respect to the measurement points, also the probe lengths of the two WindScanner systems are different (see Table 4). What effects the dimension and position of the intersecting probes have on the retrieved wind speed information will be studied in future publications.

## 5.2 Improving the methodology

In this paper, we presented the methodology for conducting field studies with multi-Doppler lidars. This was a preliminary attempt to outline and define systematic steps that can lead to the acquisition of high-quality datasets from field studies. The application of the methodology on the Perdigão 2015 helped in devising future improvements.

Simulation results could be used to assess whether WindScanners can capture the desired flow features by basically simulating the flow study itself. This calls for the integration of an end-to-end WindScanner measurement process simulator with a flow model (e.g., Vasiljević et al., 2011). The simulation of flow study is useful for selecting an appropriate type of WindScanner (pulsed and/or CW lidar) and a number of WindScanner units, and designing and optimizing scanning strategies that can capture desired flow features.

As the WindScanner wind vector retrieval accuracy is influenced by the LOS uncertainty, pointing accuracy, and retrieval technique, extending the simulator with the lidar measurements uncertainty model would provide grounds for a preliminary accuracy assessment and an optimization of the WindScanner installation locations (see accuracy maps in Vasiljević and Courtney, 2017) . In order to track the LOS accuracy, the WindScanner measurements should be regularly checked during the field study. This can be done by employing a simple and not necessarily tall mast on the field study site and installing as a minimum one sonic anemometer on the mast top. The mast top should be visible from every WindScanner location, and preferably the mast top location should coincide with one of the measurement points.

To better assess pointing accuracy more than one hard target should be used and mapped prior, during, and after the field study. The number of hard targets could correspond to the number of error sources (e.g., leveling, mirror alignment, etc.), since this would allow calculating the coordinate system for steering the laser beams that can compensate for pointing errors (see pg. 105, Vasiljević, 2014b). Mapping hard targets in several instances during the field study would provide means to update the compensating coordinate system.

The methodology for atmospheric multi-Doppler lidar experiments was applied in the pilot study Perdigão-2015, which serves as an introductory campaign to more extensive and longer planned field studies within the NEWA project. In this field campaign, the long-range and short-range WindScanner systems were used simultaneously for the first time. Also, it was the first time that these instruments were deployed in a challenging site. Over a short period of time, both systems were installed at the designated locations. The established WindScanner calibration and configuration procedures, which were previously used in flat terrain, were successfully applied in the heavily complex terrain of Serra do Perdigão. A pointing accuracy of $0.05^{\circ}$, and a high level of the synchronization among WindScanners was achieved. Also, two long-range WindScanners on the North ridge were powered by diesel generators without interruptions during their operation.

An important step has been made towards coupling of the long-range and short-range WindScanner systems into a hybrid WindScanner system. Namely, the scanning strategies were designed and implemented in such a way as to achieve a symbiosis of the two WindScanner systems. However, in this first attempt to realize a hybrid WindScanner system, we can report that we managed to simultaneously operate a hybrid WindScanner system for several hours. A longer operation was hindered by the issues earlier mentioned in Section 4.8. The coinciding measurement periods will be throughly addressed in forthcoming

publications. An obvious focus of future campaigns with a hybrid WindScanner system is to achieve a full synergy between
the long- and short- range WindScanner systems, thus to acquire longer simultaneous measurements of flow fields.

Despite the environmental conditions, challenges imposed by the site, and technical issues with the instruments, high-quality
observations of the various flow aspects of the Perdigão site have been acquired, with a few highlights outlined in this paper.
Also, the datasets have been used for the design of a longer, and instrumentation-wise more extensive Perdigão-2017 field
campaign, that is taking place in the first half of 2017 over a period of six months.

# 6    Conclusion

The methodology for atmospheric multi-Doppler lidar experiments was developed and applied to the Perdigão-2015 field
campaign. We described the 10 steps which constitute the methodology and explained how each step was implemented in
the field campaign. The application of the steps resulted in a high pointing accuracy and temporal synchronization of the
WindScanners. Five novel scanning modes were designed, three in the case of the long-range, and two in the case of the
short-range WindScanner system, with the purpose of characterizing the overall flow pattern over the double-ridge site. Each
scanning mode followed a specific purpose, disclosing a particular aspect of the flow. Because the larger area was covered
by the WindScanner measurements compared to standard tower based anemometry, a more detailed view of the atmospheric
flow was possible, which increased our understanding of the interplay between the large synoptic scales associated with the
weather conditions and the site. An important step was made towards the realization of a hybrid WindScanner system, based on
both pulsed and CW lidar technology. Overall, the methodology and its application layout foundations for much larger future
endeavors that will take place within the NEWA project.

## Appendix A:  Wind vector retrieval

Based on the meteorological convention, the wind vector is defined by the three velocity components:

$$V_{wind} = (u, v, w) \tag{A1}$$

where, $u$ is the zonal velocity (i.e., component of the horizontal wind towards East), $v$ is the meridional velocity (i.e.,
component of the horizontal wind towards North), and $w$ is the vertical velocity, which is positive for an upward motion.

The LOS or radial speed, $v_{LOS}$, measured by a lidar represents a projection of the wind vector $V_{wind}$ on the laser light
propagation path:

$$v_{LOS} = n.V_{wind} = \begin{pmatrix} sin(\theta)cos(\varphi) \\ cos(\theta)cos(\varphi) \\ sin(\varphi) \end{pmatrix} \cdot \begin{pmatrix} u \\ v \\ w \end{pmatrix} \tag{A2}$$

where, $n$ is a unit vector describing the direction of the laser light propagation expressed in terms of the azimuth angle $\theta$ and
elevation angle $\varphi$.

By measuring three independent radial velocities ($v_{LOS1}$,$v_{LOS2}$ and $v_{LOS3}$ ) we can retrieve (triple-Doppler retrieval) all three components of the wind vector:

$$
\begin{pmatrix} u \\ v \\ w \end{pmatrix} = \begin{pmatrix} sin(\theta_1)cos(\varphi_1) & cos(\theta_1)cos(\varphi_1) & sin(\varphi_1) \\ sin(\theta_2)cos(\varphi_2) & cos(\theta_2)cos(\varphi_2) & sin(\varphi_2) \\ sin(\theta_3)cos(\varphi_3) & cos(\theta_3)cos(\varphi_3) & sin(\varphi_3) \end{pmatrix} \cdot \begin{pmatrix} v_{LOS1} \\ v_{LOS2} \\ v_{LOS3} \end{pmatrix}
\tag{A3}
$$

If the elevation angle is low (e.g., $\varphi < 5^{\circ}$) and vertical velocity is low (e.g., $< 2$ m/s) then Equation A2 can be reduced to:

$$
v_{LOS} = \begin{pmatrix} u \\ v \end{pmatrix} \cdot \begin{pmatrix} sin(\theta) \\ cos(\theta) \end{pmatrix}
\tag{A4}
$$

since:

$$
cos(5^{\circ}) = 0.996
\tag{A5}
$$
$$
sin(5^{\circ}) = 0.087
\tag{A6}
$$
$$
\tag{A7}
$$

Therefore, the radial velocity can be treated as the projection of the horizontal components of the wind vector on the laser light propagation path. Accordingly, by measuring two independent radial velocities (dual-Doppler retrieval) with laser beams directed at low elevation angles the horizontal components of the wind vector can be retrieved:

$$
\begin{pmatrix} u \\ v \end{pmatrix} = \begin{pmatrix} sin(\theta_1)cos(\varphi_1) & cos(\theta_1)cos(\varphi_1) \\ sin(\theta_2)cos(\varphi_2) & cos(\theta_2)cos(\varphi_2) \end{pmatrix} \cdot \begin{pmatrix} v_{LOS1} \\ v_{LOS2} \end{pmatrix}
\tag{A8}
$$

*Acknowledgements.* The authors are grateful to the following institutions, without whom the Perdigão-2015 field campaign would have not been possible. The municipality of Vila Velha de Ródão was extremely useful and always available to answer all our requests, providing solutions to many simple and practical aspects of the experiment that otherwise would have been insurmountable difficulties. We thank Generg for collaboration and support of the field experiment, namely providing power and data connection and for the availability of the wind turbine SCADA data. We thank the Portuguese Institute for Sea and Atmosphere (IPMA) for the daily forecasts. We also thank Regacentro Comércio de Representações Lda for the ingenious solution that enabled off-grid power to two scanning lidars in remote locations. Particularly, we would like to express a special gratitude to Per Hansen and Claus Pedersen from DTU Wind Energy for their dedicated work and positive spirit during the experiment deployment and execution. We are grateful to the FarmOpt, UniTTe, NEWA and WindScanner.eu projects by the

financial support of the Perdigão-2015 field campaign. FarmOpt was funded by the Danish Energy Technology Development and Demonstration Program (EUDP), Project No. 64013-0405. UniTTe is financially supported by the Danish Innovation Fund (Innovationsfonden), Grant No. 1305-00024A. NEWA is an ERANET+ project, which is funded by the European Commission (ENER/FP7/618122/NEWA) and 9 national funding agencies. WindScanner.eu was funded by European Commission under the FP7-INFRASTRUCTURES call, Project no. 312372. Finally, the first author would like to express gratitude to the IRPWind mobility program which provided a financial support for his stay in Portugal during the Perdigão-2015 field campaign.

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
