# Peer review of "Perdigão 2015: methodology for atmospheric multi-Doppler lidar experiments"

_Atmospheric Measurement Techniques, 2017_

## Short Comment (SC1) · 1 Mar 2017

Nikola et al.,

Very nice paper and well structured. The wake results are very interesting.

A few comments on the paper, which I believe would be good address or discuss here

1. Fig 13, is it from Long-range wind scanner or short-range wind scanner? Its not very clear in your paper. If from short-range wind scanner, what is the effect of dynamic focusing on the velocity resolution and accuracy? Can you please refer to some work done someone in your group?

2. Fig 11 & Fig 14, based on the location of your other Windscanners, the subtended angle "looks" small and maybe below 50 deg, is my assessment correct? You could

compare it with a sonic/tower at one of the location, maybe that would help validate your results. And also would be good to make note of the subtended angle between the two beams at the point of measurement.

3. What has been your average range observed by the windscanners at Perdigao?

4. Regarding the filtering criteria in table 5 - Say you have 3 Lidars, and data only from 2 Lidars is available at the location of measurement at a given time. Do you ignore all these measurements to get u, v & w? Or do you just calculate the 2 components (u & v) and the third (w) is a NaN? It would be good to test this out, based on the subtended angle between the two Lidars at the measurement location and elevation angle, maybe?

———————————————————

---

## Referee Comment (RC1) · Anonymous Referee #1 · 6 Mar 2017

The manuscript presents a methodology for deriving wind speeds over large and small scales from a combination of windscanners. Recognizing the enormous effort this takes, the methodology is clearly explained here. Perhaps a point that is slightly less clear is the complexity of analyzing and presenting data from these large datasets. I don't think there is room for this also in this manuscript and hope it will be discussed in later work by these authors. The novel aspects of the paper are in the detailed methodology. A major omission lies in the discussion of the retrieval of wind field from the short and long-range scanners and the issues associated with each in complex terrain (pulse vs cw). This might just need a citation but it is an important issue. Another point that is not completely clear is how good or otherwise the agreement is between the wind fields derived from two scanners? Understanding this was only possible for a few hours – is it possible to plot the two locations/wind speeds together? This would

show a major step forward in the coupling both in terms of the physical (where did the scanners overlap and with what frequency) and the data comparison (i.e. do the two derived wind speeds agree in space). Did I miss the discussion of the issues of lidar operation in high temperatures ?– these might be really useful operationally. Figure 3. Please indicate the purpose of the shading. Figure 4. Please indicate the meaning of the thick lines. Please give a reference for the coordinate system (p4, l73). There are a few minor typos please check for those. Otherwise it is a useful contribution on a major innovation that can be published subject to these minor issues.

―――――――――――――――――

---

## Referee Comment (RC2) · Anonymous Referee #2 · 24 Mar 2017

This manuscript is about multiple lidar measurements performed for the Perdigao 2015 campaign with two triple-Doppler units, both pulsed long-range lidars and continuous short-range lidars.

The first criticism is about the writing style. Rather than a scientist paper, this document reads like a romantic technical report, or maybe a long post on a blog. The manuscript is very very lengthy. The first lidar data is shown at page 20 (the last one at page 23). If I am not mistaken, no data from the short-range system is provided. I would rather recommend a more classical structure of the manuscript consisting of introduction, description of the site and setup, lidar scans and data retrieval, discussion of the results and conclusion.

Besides the writing, I have also some concerns about the novel results presented in this

manuscript. We have seen already dual-Doppler lidar measurements in wind turbine wakes or vertical transects for valley flows. Two figures are definitely not sufficient to describe effects of atmospheric stability on wind turbine wakes. Therefore, I suggest to provide a sharper focus on the data analysis and emphasize any new result.

Comments are provided below for a revision of the manuscript.

Comments: 1. P1L3: "...measure mean flow conditions over an entire region...", this sounds a bit too vague, maybe better stating the typical measurement volume of the two systems. 2. P2L3: "...it is unrealistic to sample...", actually it is real performing met-tower measurements. Maybe it is better mentioning the reasons why a multi-lidar system can be advantageous. 3. Sect. 1: It seems to me that this introduction is lacking to provide an overview of existing works on triple lidar measurements, such as J. Mann et al. 2009, Meteor. Z. 18, 135-140, Fuertes et al. 2014, JTECH 31(7), 1549-1556, or papers from the AMT special issue on the XPIA experiment (http://www.atmos-meas-tech.net/special_issue645.html), which was focused on assessing various multiple-lidar scanning strategies (Lundquist et al. 2017, BAMS 98(2), 289-314). Therefore, I suggest providing a more comprehensive introduction on the topic. 4. P3L7: Provide some references for the setup of the SRWS. 5. P3L14: Discuss the motivations on developing a hybrid system. 6. Sect. 3: these 10 steps are common for any (field) experiment and not specifically related to the LRWS and SRWS. Why these steps should have a special relevance or being different for this experiment? 7. Sect. 4 and throughout the paper: I understand the passion and excitement of the authors; however, this writing style is more adequate for a blog or a newspaper article rather than a scientific paper. Comments like "need to test both the equipment and human resources in highly demanding field experiments (P4L15)", "harsh conditions, high temperature and remote locations" ... This experiment was carried out in Portugal, I cannot image what scientists in Antarctica should write to describe their experiments! 8. Sect. 4.1-Sect. 4.4. This description is lengthy and unfocused. It would be easier to provide a classical description of the site and instrumentation. 9. Sect. 4.4: You suddenly introduce these

unexpected RANS simulations over the topography, without canopy performed with a commercial code, and finally just saying that the "...high complexity of the flow (Fig. 4) and large recirculation zone enclosed in the valley (Fig. 5)". I think these comments were highly expectable. Thus, I suggest removing the entire paragraph on the simulations and Figures 4 and 5. 10. P8L20: Maybe can be a personal issue of this reviewer, there is no way I can remember the names of these 3 lidars. I suggest to name them as LR1, LR2 and LR3 rather than with your nick names. 11. P8L22: What do you mean for "entailing the wind turbine"? Maybe a vertical plane along the line connecting a lidar with the turbine? 12. P8L25-29: Maybe add another table with distances among the different objects. 13. Sect. 4.5: this section can be completely removed. It provides only unimportant information. 14. P12L30-P14L3: If the measurement plane of this dual Doppler lidar was inclined, how is it possible you retrieved horizontal wind speed and direction? I guess you retrieve the 2 velocity components over the measurement plane. 15. P15L6-L11: The description of this scan is very confusing. You say that there was a time delay among the different lidars, and the delay was increasing with time. Therefore, you need to provide a statistical characterization of this delay and how you treat this time delay in the retrieval of the wind velocity components. 16. Sect. 4.7: A general comment for all the presented scanning strategies: for multiple Doppler lidar measurements, accuracy in the retrieval of the wind velocity components is affected by the elevation and azimuthal angles of the various lidars. A criterion for quantifying this error over a scan has been proposed in Debnath et al. 2017 AMT, 10, 431-444. A similar analysis should be provided for the proposed scans. 17. Sect. 4.7: A general comment for all the presented scanning strategies: you haven't provided any information on the data retrieval of wind velocity components from the lidar radial velocities. This part should be included in the manuscript. 18. Sects. 8 and 9 can be removed or summarized in the description of the setup. 19. Sect. 4.10 provides should significantly shortened. 20. P19L9: what do you mean for ... "show no turning of the wind for Northeast winds...divergency of flow lones"? This does not sound like a technical language. 21. Fig. 11 is not described accurately in the text. Why you were not able

to get measurements in the turbine wake? 22. Figs 12 and 13. You should provide more information about the wind condition, day, atmospheric stability, etc. No details are provided on the data retrieval. 23. P20L1-2: "The inflow and wake of the turbine during a one full day is well represented in Vasiljevic ÌĄ (2016b)". Why then you provide these figures if a deeper analysis has been already published?

---

## Author Comment (AC1) · 10 May 2017

**Note**: The structure of the replies to the reviewer comments is as following: (1) the original unchanged reviewer comments are given with regular text formatting , (2) the comments are enumerated and type of comment is indicated (general comment or specific comment), (3) following each comment a response to the comment and description of associated changes in the revised manuscript are provided while the text is formatted *italic*.

**General comment 1**: Very nice paper and well structured. The wake results are very interesting.

*Dear Reviewer,*
*We would like to thank you for your time and for your insightful comments which were*

*used to revise and improve our manuscript.*
*Find our responses below.*

**Specific comment 1**: Fig 13, is it from Long-range wind scanner or short-range wind scanner? Its not very clear in your paper. If from short-range wind scanner, what is the effect of dynamic focusing on the velocity resolution and accuracy? Can you please refer to some work done someone in your group?

*Figure 13 displays measurements acquired by the short-range WindScanner system. In the revised manuscript we have updated Figure 13 caption to indicate this. Also, captions of other figures that show observational highlights (i.e., Figure 11 - 15) are updated in a similar way. To our knowledge, the dynamic focusing does not impact the velocity resolution and accuracy since the number of FFT points used to perform the spectral analysis remains unchanged from one to another focusing state. In general, the focusing in CW lidars has implication on the probe length. The closer we focus the CW laser beam the smaller is the probe length and further we focus the beam the bigger is the probe length. Therefore, in Table 4 one can notice that we provided not a single number but a range of values for the probe length of the short-range WindScanner system that is the result of various focusing states throughout T and Vertical scans.*

**Specific comment 2**: Fig 11 & Fig 14, based on the location of your other Windscanners, the subtended angle "looks" small and maybe below 50 deg, is my assessment correct? You could compare it with a sonic/tower at one of the location, maybe that would help validate your results. And also would be good to make note of the subtended angle between the two beams at the point of measurement.

*Instead of the subtended angle let us expresses the geometry of a multi-Doppler laser beam steering in terms of the intersecting and elevation angles. The intersecting*

*angle we define as the smallest angle between the projections of two intersecting laser beams in a horizontal plane. The intersecting angle can take any value between 0 and 90 degrees. If the elevation angle is zero the intersecting angle is equal to the subtended angle.*

*When setting up the layout of a multi-lidar experiment we intend to have an intersecting angle of at least 30 degrees respect to the prevailing wind direction. Based on our simple accuracy model (see Vasiljević, N. and Courtney, M. Accuracy of dual-Doppler lidar retrievals of near-shore winds, 2017, WindEurope Resource Assessment Workshop 2017, https://goo.gl/LFuimU) the intersecting angle of 30 degrees results in the accuracy of about 0.25 m/s for the retrieved horizontal wind speed.*

*Following this rule of thumb and in connection to the prevailing wind directions (Northeast and Southwest) we design the layout of the Perdigão. Accordingly, throughout the Diamond scan on average the intersecting angle was 49.37 degrees, while for the Ridge scan it was 42.21 degrees. The prevailing flow was such that it was basically going through the middle of the intersecting angle, resulting in a good projection of the wind vector on the lidars line-of-sights. For the both scanning methods the elevation angles at which the laser beams were steered were about 4 degrees. Thus, the beams were almost steered in a horizontal plane.*

*Since for the Transect, T and Vertical scan the WindScanner systems were run in a triple-Doppler mode there is not a single value for the intersecting angle but three values. For the Transect scan the intersecting angle between Koshava and Sterenn, Sterenn and Whittle and Koshava and Whittle were 84.10 degrees, 54.77 degrees and 41.12 degrees respectively. They remained constant during the scan. In case of the T-scan, the mean intersection angles between R2D1 and R2D2, R2D1 and R2D3, and R2D2 and R2D3 were 44.26 degrees, 55.78 degrees and 49.64 degrees respectively. Finally, the mean intersection angles between R2D1 and R2D2, R2D1 and R2D3, and*

[Figure]

*R2D2 and R2D3 in the case of the Vertical scanning mode were 28.77 degrees, 59.96 degrees and 49.21 degrees respectively.*

*In the revised manuscript, the mean intersecting and elevation angles are now indicated throughout Section 4.7.*

*Our intention was to have a functional met mast installed at site for a continuous sanity check of WindScanners measurements during the campaign. Due to technical difficulties, this was not achieved for 2015 edition of the Perdigão experiment. However, prior and after the Perdigão-2015 we validated WindScanner measurements against a mast at our test site in Denmark. A report of the post-Perdigão validation, which was a pre-RUNE validation campaign (see Floors et al. 2016) for several long-range WindScanners is publically available (see https://goo.gl/FUuPGv).*

*For the Perdigão-2017 experiment, we followed the recommendations outlined in Section 5.2 (specifically line 16 to 22) of the reviewed manuscript. In accordance with these recommendations, we designed the scanning methods in such way that the location of at least one measurement point of a scanning method coincides with the location of one mast-based sensor. This will allow the validation of the multi-lidar measurements during the Perdigão-2017 experiment.*

**Specific comment 3**: What has been your average range observed by the windscanners at Perdigão?

*The average range of the long-range WindScanners was approximately 2.3 km. However, it should be noted that the maximum range we configured was 2.5 km. Therefore, it is hard to judge what would be the actual average range of a pulsed lidar for the Perdigão site.*

**Specific comment 4**: Regarding the filtering criteria in table 5 - Say you have 3 Lidars, and data only from 2 Lidars is available at the location of measurement at a given time. Do you ignore all these measurements to get u, v & w? Or do you just calculate the 2 components (u & v) and the third (w) is a NaN? It would be good to test this out, based on the subtended angle between the two Lidars at the measurement location and elevation angle, maybe?

*For T and Vertical scans, we provided reconstructed u, v and w components when all three LOS measurements were available. The exception is made for the Transect scan and only in the case when Whittle measurements are 'NaN'. In this special case, the horizontal wind speed is reconstructed from Sterenn and Koshava since the intersecting angle between the laser beams from these two WindScanners is close to 90 degrees. In case of the Diamond and Ridge scans we only reconstructed u and v components when measurements from both Sterenn and Koshava were available.*

---

## Author Comment (AC2) · 10 May 2017

**Note**: The structure of the replies to the reviewer comments is as following: (1) the original unchanged reviewer comments are given with regular text formatting , (2) the comments are enumerated and type of comment is indicated (general comment or specific comment), (3) following each comment a response to the comment and description of associated changes in the revised manuscript are provided while the text is formatted *italic*.

**General comment 1**: The manuscript presents a methodology for deriving wind speeds over large and small scales from a combination of windscanners. Recognizing the enormous effort this takes, the methodology is clearly explained here. Perhaps a point that is slightly less clear is the complexity of analysing and presenting data from

these large datasets. I don't think there is room for this also in this manuscript and hope it will be discussed in later work by these authors. The novel aspects of the paper are in the detailed methodology.

*Dear Reviewer,*
*We appreciate the time you dedicated for reviewing the manuscript and for providing constructive comments. Especially we would like to thank you for recognizing the efforts that have been put to produce the presented results (methodology and experiment). Our replies follow.*

*We agree that we did not show in many details the complexity of analyzing and presenting data from the acquired datasets. We did touch this topic in Section 4.10. We could only add to the reviewer comment that the lidar data topic is rarely presented in publications. We are in the preparation of a manuscript specifically addressing the lidar data topic wherein more details we will present the lidar data complexity.*

**Specific comment 1**: A major omission lies in the discussion of the retrieval of wind field from the short and long-range scanners and the issues associated with each in complex terrain (pulse vs cw). This might just need a citation but it is an important issue.

*We revised our manuscript in accordance with the referee's comment. In the revised manuscript Section 1 is extended and includes a discussion and references on single lidar errors in complex terrain and historical overview of multi-Doppler efforts. Furthermore, we extended Section 2 which now includes clearer motivations for developing a hybrid WindScanner system.*

**Specific comment 2**: Another point that is not completely clear is how good or otherwise the agreement is between the wind fields derived from two scanners? Understanding this was only possible for a few hours – is it possible to plot the two

locations/wind speeds together? This would show a major step forward in the coupling both in terms of the physical (where did the scanners overlap and with what frequency) and the data comparison (i.e. do the two derived wind speeds agree in space).

*The manuscript has been updated with a figure displaying simultaneous measurements (short- and long- range) of the wind turbine inflow conditions for one 10-minute period. Also we have provided a plot showing the intercompatison of the short- and long- range WindScanner measurements during this period. At the conciding measurement points the retrieved horizontal wind speed by the long- and short- range WindScanner system show a good agreement (averaged difference 0.2 m/s). Detailed analysis of the simulatenous observation periods, particularly respect to the positions and dimensions of the intersecting probe volumes will be addressed in a separet publications, since the current publication is already lengthy and generally speaking more focused on the methodlogy how to do a multi-lidar experiment.*

**Specific comment 3**: : Did I miss the discussion of the issues of lidar operation in high temperatures? – these might be really useful operationally.

*The issues with high temperatures have been briefly discussed at Page 17 Line 18-20 in the reviewed manuscript. In the revised manuscript, a reference has been added indicating possible issues with the cooling system of the long-range WindScanner system and potential solutions for the future use of the long-range WindScanner system in warm climates.*

**Specific comment 4**: Figure 3. Please indicate the purpose of the shading. Figure 4. Please indicate the meaning of the thick lines. Please give a reference for the coordinate system (p4, l73). There are a few minor typos please check for those. Otherwise it is a useful contribution on a major innovation that can be published subject to these minor issues.

*In the revised manuscript, the shading in Figure 3 has been removed from the plots as it does not add any additional information to the figure. The thick lines in Figure 4 represent the South ridge, valley and North ridge line. These lines are now denoted in the revised version of Figure 4. We updated the manuscript with the reference for the ETRS89 coordinate system. Furthremore, the revised manuscript has been proofread and corrected by a native English speaker.*

---

## Author Comment (AC3) · 10 May 2017

**Note**: The structure of the replies to the reviewer comments is as following: (1) the original unchanged reviewer comments are given with regular text formatting , (2) the comments are enumerated and type of comment is indicated (general comment or specific comment), (3) following each comment a response to the comment and description of associated changes in the revised manuscript are provided while the text is formatted *italic*.

**General comment 1**: This manuscript is about multiple lidar measurements performed for the Perdigão 2015 campaign with two triple-Doppler units, both pulsed long-range lidars and continuous short-range lidars.

*Dear Reviewer,*
*We would like to thank you for all the care and attention given to our manuscript. Our answers are given below.*

*We partially agree with the general review remarks regarding our manuscript. In fact, the manuscript is about the methodology for multi-Doppler lidar experiments in the Perdigão-2015 field campaign, as evidenced by the title. Data analyses or discussions of particular flow situations are not the purpose of the present manuscript; the measurements are included only as a result of the presented methodology.*

**General comment 2**: The first criticism is about the writing style. Rather than a scientist paper, this document reads like a romantic technical report, or maybe a long post on a blog. The manuscript is very very lengthy. The first lidar data is shown at page 20 (the last one at page 23). If I am not mistaken, no data from the short-range system is provided. I would rather recommend a more classical structure of the manuscript consisting of introduction, description of the site and setup, lidar scans and data retrieval, discussion of the results and conclusion.

*The writing style is not a major issue in scientific publications, as opposed to contents, structure or accuracy. See for instance, that the structure of our manuscript is rather formal; i.e. IMRAD (Introduction-Methodology-Results-and-Discussion), as recommended by the referee.*

*This is a manuscript in the first place about the methodology for multi-lidar experiments, but also about a field experiment, the first of its kind. This experiment has been made with many difficulties where many of them had to be fixed while running the experiment.*

*We wanted to report our difficulties, even when the solutions failed, for the ben-*

*efit of future field experiments. It is not a typical, standard, scientific paper, where difficulties and things that did not work as expected are ignored, because are considered irrelevant.*

*Science is made of decisions that in the end proof to be wrong and equipment that fails to work under more demanding conditions. This could not be achieved in the style of a standard scientific paper.*

*Science is also made by people, most of them are passionate about their work and enjoy the challenge of overcoming difficulties. We wanted that to transpire in our text, and we are glad we made it, though to the dissatisfaction of the referee, by not complying with the commonest and traditional papers of the last 20 years. We cherish the writing style and recommendations of excellent scientists and texts such as:*

*– Advice for a Young Investigator, S. Ramon y Cajal. 2004 new edition. Unabridged and unaltered reproduction of the first edition, published in 1897. 172 pages.*

*– On Being a Scientist: Responsible Conduct in Research. National Academy Press, 1995, 40 pages*

*– How to Write and Publish a Scientific Paper (2016) Cambridge University Press, Robert A. Day and Barbara Gastel. 8th edition. 350 pages*

*We like the clarity, the thoroughness and the simplicity of good classical scientific papers that manage to describe the work and the ingenuity of the experiments, and also pass the personal traits of their authors. Characteristics that are difficult to find in today's papers.*

*This manuscript is about the methodology for multi-Doppler lidar experiments.*

*We did indeed report on the experiment and acquired dataset and it is not much different from (when it comes to its content and form) when compared to several publications reporting conducted experiments without in-depth data analysis. See for instance:*

*– Grubisić et al. 2008 on the T-Rex campaign*

*– Floors et al. 2016 on the RUNE experiment*

*The short-range WindScanner data are presented in Figure 13 of the reviewed manuscript. We updated the captions of Figures 11 to 14 to indicate whether data is acquired with the long- or short- range WindScanner system.*

**General comment 3**: Besides the writing, I have also some concerns about the novel results presented in this manuscript. We have seen already dual-Doppler lidar measurements in wind turbine wakes or vertical transects for valley flows. Two figures are definitely not sufficient to describe effects of atmospheric stability on wind turbine wakes. Therefore, I suggest to provide a sharper focus on the data analysis and emphasize any new result.

*Regarding the novelty of the manuscript, the following is a list summarizing novel contributions:*

1. *Methodology for atmospheric multi-Doppler lidar experiments*

2. *Novel scanning methods, such as for example T-scan, ridge scan and diamond scan*

3. *Use of two different (CW and pulsed based) multi-Doppler lidar systems simultaneously*

4. *Dual-Doppler and triple-Doppler measurements of a single turbine wake and in-flow conditions in complex terrain*

5. *Wind resource measurements along a ridge*

6. *Mapping valley flows over a vertical transect (novel since it was done for a double-hill site)*

*Detailed data analysis of several aspects of the Perdigão flow has been presented in several communications (e.g., Rodrigues et al. 2016 and Hansen et al. 2016), which are referenced in our paper (see P19L12 – L14 in the reviewed manuscript). The atmospheric stability can and was addressed as a hypothesis, because there were no temperature or heat flux measurements. The impact of the atmospheric stability on wind turbine wake has been partially discussed in Hansen et al. 2016.*

**Specific comment 1**: P1L3: "...measure mean flow conditions over an entire region...", this sounds a bit too vague, maybe better stating the typical measurement volume of the two systems.

*Following the referee's suggestion, the sentence has been modified. It reads:*

*"Due to the costs of tall meteorological masts, especially in complex terrain, it is unrealistic to sample the wind within an entire region occupied by today's largest wind turbines or farms with traditional anemometry."*

**Specific comment 2**: P2L3: "...it is unrealistic to sample...", actually it is real performing met-tower measurements. Maybe it is better mentioning the reasons why a multi-lidar system can be advantageous.

*Following the referee's suggestion, the paragraph have been revised. It reads:*

*"Due to the costs of tall meteorological masts, especially in complex terrain, it is unrealistic to sample the wind within an entire region occupied by today's largest wind turbines or farms with traditional anemometry. This is exactly what can be achieved with multi-lidar systems."*

**Specific comment 3**: Sect. 1: It seems to me that this introduction is lacking to provide an overview of existing works on triple lidar measurements, such as J. Mann et al. 2009, Meteor. Z. 18, 135-140, Fuertes et al. 2014, JTECH 31(7), 1549-1556, or papers from the AMT special issue on the XPIA experiment (http://www.atmos-meas-tech.net/special_issue645.html), which was focused on assessing various multiple-lidar scanning strategies (Lundquist et al. 2017, BAMS 98(2), 289-314). Therefore, I suggest providing a more comprehensive introduction on the topic.

*We agree with the reviewer that an overview of multi-lidar experiments is missing in the reviewed manuscript. Therefore, the revised manuscript includes an overview of multi-lidar efforts among which the suggested references are included.*

**Specific comment 4**: P3L7: Provide some references for the setup of the SRWS.

*We updated manuscript with additional references to the short-range WindScanner system (Mikkelsen et al. (2011) and Sjoholm et al. (2014)).*

**Specific comment 5**: P3L14: Discuss the motivations on developing a hybrid system.

*The revised version was changed in accordance to the referee's suggestion. Now it includes clearer motivations for developing a hybrid WindScanner system.*

**Specific comment 6**: Sect. 3: these 10 steps are common for any (field) experiment and not specifically related to the LRWS and SRWS. Why these steps should have a special relevance or being different for this experiment?

*The 10 steps are our proposal to a systematic approach to future multi-lidars (WindScanners) campaigns. The classification, ordering of the many activities from the beginning until the end of the campaign under these 10 steps appeared logical to us and was most useful while preparing the currently ongoing, much larger field campaign Perdigão-2017.*

*Yes, certainly there are steps which are common to any experiment, but there are steps which are specific for scanning lidars; for instance, the scanning patterns design.*

*We felt the need for the systemization of WindScanner campaings within the ESFRI European Infrastructure project where the main purpose is setting a distributed and mobile infrastructure for these type of activities, which we also believe will be useful when organizing future large field experiments.*

**Specific comment 7**: Sect. 4 and throughout the paper: I understand the passion and excitement of the authors; however, this writing style is more adequate for a blog or a newspaper article rather than a scientific paper. Comments like "need to test both the equipment and human resources in highly demanding field experiments (P4L15)", "harsh conditions, high temperature and remote locations"... This experiment was carried out in Portugal, I cannot image what scientists in Antarctica should write to describe their experiments!

*See above the reply to the general comment 2.*

**Specific comment 8**: Sect. 4.1-Sect. 4.4. This description is lengthy and un-focused. It would be easier to provide a classical description of the site and instrumentation.

*Field experiments are very demanding in both human and financing resources, Perdigão-2015 was possible and justified by four independent projects (WindScanner.eu, NEWA, UniTTe and FarmOpt), which resulted in collecting experimental and unique datasets focused in the characterization of the wind turbine wake, the flow separation on lee sides of hills (also the valley flow and recirculation zone), etc.. Rigour in the description of the thinking, which was behind the organization and preparation of the field campaign, led to this section that seems lengthy in a first reading.*

**Specific comment 9**: Sect. 4.4: You suddenly introduce these unexpected RANS simulations over the topography, without canopy performed with a commercial code, and finally just saying that the "...high complexity of the flow (Fig. 4) and large recirculation zone enclosed in the valley (Fig. 5)". I think these comments were highly expectable. Thus, I suggest removing the entire paragraph on the simulations and Figures 4 and 5.

*The RANS simulations were extremely useful by guiding us in the positioning of the lidar units. The CFD code is not a commercial code and the presence or not of the canopies does not change the main features of the flow. The use of computer modelling prior to field campaigns is a must and a practice that will become more and more common. The self-imposed limitations in the extent of the manuscript did not allow for more detailed analysis of the flow pattern and that was left to the reader. The intricacies of the separated flow, namely in the lee sides, could not be foreseen without flow modelling.*

**Specific comment 10**: P8L20: Maybe can be a personal issue of this reviewer,

there is no way I can remember the names of these 3 lidars. I suggest to name them as LR1, LR2 and LR3 rather than with your nick names.

*As suggested by the referee, the identifications ends with a number:*

- *Koshava – LR1*

- *Sterenn – LR2*

- *Whittle – LR3*

- *R2D1 – SR1*

- *R2D2 – SR2*

- *R2D3 – SR3*

*The names are part of the identification, to track the usage in field operations and for consistency with previous publications; for instance:*

*Floors, Rogier and Peña, Alfredo and Lea, Guillaume and Vasiljević, Nikola and Simon, Elliot and Courtney, Michael (2016). The RUNE Experiment–A Database of Remote-Sensing Observations of Near-Shore Winds. Remote Sensing, V. 8, page 884, N. 11, doi = 10.3390/rs8110884, url = http://www.mdpi.com/2072-4292/8/11/884*

**Specific comment 11**: P8L22: What do you mean for "entailing the wind turbine"? Maybe a vertical plane along the line connecting a lidar with the turbine?

*The quoted sentence was modified, accordingly to the referee's suggestion.*

**Specific comment 12**: P8L25-29: Maybe add another table with distances among the

different objects.

*Following the referee's suggestion, we added two additional columns to Table 3 to show distance and direction of lidars and landmarks of interest respect to the wind turbine position. Also, we denoted WindScanners with short names in Figure 6.*

**Specific comment 13**: Sect. 4.5: this section can be completely removed. It provides only unimportant information.

*The manuscript was organized in such a way that each step in the procedure has a corresponding section. Steps (or sections) are the result of organizing (under a logical structure) a large number of activities. The number and ordering of steps took into consideration many aspects, including the importance and the number and relationships among the activities within each step.*

*Section 4.5 (step 5) is on infrastructures (power and data network, access roads, etc.) and infrastructures are independent, different and autonomous from other aspects of a field campaign that must be considered when planning an experiment. The infrastructure complies with all features to make it a step in the methodology for lidar experiments.*

**Specific comment 14**: P12L30-P14L3: If the measurement plane of this dual Doppler lidar was inclined, how is it possible you retrieved horizontal wind speed and direction? I guess you retrieve the 2 velocity components over the measurement plane.

*We agree with the referee. The two components were retrieved in the inclined plane. However, because the elevation angle was low these components are close to the components resolved in an absolute horizontal plane. The revised manuscript has been adequately updated to reflect the previous statements and additional calculations*
*are provided in Appendix A of the revised manuscript.*

**Specific comment 15**: P15L6-L11: The description of this scan is very confusing. You say that there was a time delay among the different lidars, and the delay was increasing with time. Therefore, you need to provide a statistical characterization of this delay and how you treat this time delay in the retrieval of the wind velocity components.

*The commented paragraph was rewritten and the statistical characterization of the lag was given in term of a lag rate (i.e., the speed at which the lag increase in time). In the reviewed manuscript we indicated how we treated the time delay (P18L19-L20):*

*"To treat the synchronization issues that appeared in the expanded version of T-scan and vertical plan patterns, the data acquired using those pattern versions were additionally time averaged in 10-min periods."*

**Specific comment 16**: Sect. 4.7: A general comment for all the presented scanning strategies: for multiple Doppler lidar measurements, accuracy in the retrieval of the wind velocity components is affected by the elevation and azimuthal angles of the various lidars. A criterion for quantifying this error over a scan has been proposed in Debnath et al. 2017 AMT, 10, 431-444. A similar analysis should be provided for the proposed scans.

*This is a valid point, and we updated the manuscript with rather short discussion on the expected accuracy in the retrieved wind components.*

*When setting up the layout of a multi-lidar experiment we intend to have an intersecting angle of at least 30 degrees respect to the prevailing wind direction. Based*

[Figure]

*on our simple accuracy model (see Vasiljević, N. and Courtney, M. Accuracy of dual-Doppler lidar retrievals of near-shore winds, 2017, WindEurope Resource Assessment Workshop 2017, https://goo.gl/LFuimU) the intersecting angle of 30 degrees results in the accuracy of about 0.25 m/s for the retrieved horizontal wind speed. Following this rule of thumb and in connection to the prevailing wind directions (Northeast and Southwest) we design the layout of the Perdigão-2017 experiment.*

*In the revised manuscript, we: (1) indicated the mean intersecting and elevation angles for all scanning modes, (2) referred to the aforementioned simple model, (3) indicated the accuracy for the retrieved horizontal wind speed based on this model (i.e., 0.25 m/s), and (4) referred to Simley et al., 2016 and Debnath et al. 2017 as an alternative approach for assessing the multi-lidar setup suitability.*

*It is our opinion that this topic requires a separate publication as the reviewed manuscript covers a multitude of topics. We are currently preparing several communications on the scanning lidar accuracy topic.*

**Specific comment 17**: Sect. 4.7: A general comment for all the presented scanning strategies: you haven't provided any information on the data retrieval of wind velocity components from the lidar radial velocities. This part should be included in the manuscript.

*We agree with the referee. We updated the manuscript with the necessary equations for retrieving the wind vector components from independent LOS measurements (see Appendix A in the revised manuscript).*

**Specific comment 18**: Sects. 8 and 9 can be removed or summarized in the description of the setup.

*See the response to the specific comment 13.*

**Specific comment 19**: Sect. 4.10 provides should significantly shortened.

*See the response to the specific comment 13.*

**Specific comment 20**: P19L9: what do you mean for . . . "show no turning of the wind for Northeast winds. . .divergency of flow lones"? This does not sound like a technical language.

*The paragraph has been revised. It reads:*

*"The ridge scan, 2 km along the South ridge, shows no turning of the wind for Northeast winds. Thus, for this wind direction we observed a two-dimensional flow. On the other hand, for Southwest winds there is a slight turning of the wind..."*

**Specific comment 21**: Fig. 11 is not described accurately in the text. Why you were not able to get measurements in the turbine wake?

*In Figure 11 caption it is mentioned that measurements at the wind turbine location are erroneous. This is because the laser beams were hitting the wind turbine during the ridge scan. The ridge scan was designed such that the beam intersection followed the ridge line which includes the turbine itself. Therefore, the beam intersection hits the turbine (nacelle and blades). At this location, the reported radial velocity equals the velocity of the wind turbine and not the air (see CNR mapper P12L1 to L10 for more details). Therefore, those measurements must be removed as they are not measurements of the wind but a hard target velocity.*

**Specific comment 22**: Figs 12 and 13. You should provide more information

about the wind condition, day, atmospheric stability, etc. No details are provided on the data retrieval.

*Regarding the stability see our answer to the general comment 3. Captions of Figures 11 – 14 include information on day and time. The wind conditions can be concluded from these figures.*

**Specific comment 23**: P20L1-2: "The inflow and wake of the turbine during a one full day is well represented in Vasiljevic (2016b)". Why then you provide these figures if a deeper analysis has been already published?

*The cited reference represents a link to the Youtube video that displays a 24h reconstructed wind field around the wind turbine. We followed common practices for citing web links.*

---

## Author Response (AR1)

May 10, 2017

Dear Mr. Stoffelen,

Please find the revised version of our paper entitled "Perdigão 2015: methodology for atmospheric multi-Doppler lidar experiments" to be considered for publication as a full research manuscript in Atmospheric Measurement Techniques.

All the comments from the three reviewers have been considered. A point-by-point response to the reviews are provided in the replies posted in the Interactive discussion (AMTD).

Generally, the reviewers requested a few larger revisions: (1) review of the literature covering multi-lidar experiments, (2) description of multi-lidar retrieval of the wind speed, (3) description of the accuracy of the retrieved wind speed, and (4) example of hybrid WindScanner measurements. These requests have been carefully addressed in the revised manuscript as following:

- Review of the multi-lidar literature is provided in Section 1 – Introduction (page 2 line 2 to line 22 in the revised manuscript)
- Mathematical background of the multi-lidar retrieval is provided in Appendix A
- Accuracy of the retrieved wind speed is provided in Section 4.4 (page 9 line 14 to line 26)
- Hybrid WindScanner measurements are given in Section 5.1 (page 23 line 8 to line 10, page 24 line 1 to line 6, page 25 line 1 to line 4 and Figure 15)

Other comments and feedbacks from the reviewers resulted in minor corrections and improvements of the manuscript.

To improve language, a native English speaker performed a thorough proof reading and corrections of our manuscript.

Thank you in advance for your kind consideration of this submission.

On behalf of all co-authors,

Nikola Vasiljević

---

## Author Response (AR2)

August 2, 2017

Dear Mr. Stoffelen,

We would like to thank you for the time you have spent conducting the review process and providing constructive comments. We are very pleased to read that you find our contribution acceptable for the publication in AMT and that you commend our manuscript clarity and structure.

We agree that the revised manuscript did not undergo major changes when compared to the initial submission. In fact, we extended it by adding the content that the reviewers requested (e.g., review of the multi-lidar literature, wind retrieval accuracy, etc.), and made the manuscript more clearer for future readers.

We carefully addressed your feedbacks to include some parts of the open discussion, which have not been part of the revised manuscript, in to the latest version (revision 2) of the paper. Specifically, we pointed out what is new that the manuscript brings to the readership, what assumption was made in the wake analysis, and particularly stated the wider applicability of the presented methodology. The revisions follow below.

On the page 2 we extended the paragraph which starts on the line 30:
Original paragraph: *In this paper, we propose such a methodology. The methodology will be discussed through its application to a pilot experiment that took place in a complex and forested site in Portugal.*

Revised paragraph: *In this paper, we propose such a methodology. The methodology will be discussed through its application to a pilot experiment, Perdigão-2015, that took place in a complex and forested site in Portugal, where the two WindScanner systems were operated simultaneously, for the first time. Besides the methodology and operation of the WindScanner systems, this manuscript also presents: a review of the two WindScanner systems, novel scanning methods, observational highlights of multi-lidar measurements of a single turbine wake and inflow conditions in complex terrain, multi-lidar measurements of wind resources along a ridge and observations of valley flows. It should be pointed out that the data analysis or discussions of particular flow situations are not the purpose of the present manuscript, and are included as a result and illustration of the presented methodology.*

The paragraph on the page 22 starting at the line 11 of the reviewed manuscript (wake analysis) has been clarified by adding the following statement to point out how we differentiated between different states of the atmosphere:
*Due to the lack of temperature and heat flux measurements, we established an empirical relation between the period of a day and atmospheric stability.*

In Section 5.2 (discussion) the initial paragraph has been extended to point out the applicability of the presented methodology and how the methodology can be executed differently.

Original paragraph: *In this paper, we presented the methodology for conducting field studies with multi-Doppler lidars. This was a preliminary 10 attempt to outline and define systematic steps that can lead to the acquisition of high-quality datasets from field studies.*

Revised paragraph: *In this paper, we presented the methodology for conducting field studies with multi-Doppler lidars. This was a preliminary attempt to outline and define systematic steps that can lead to the acquisition of high-quality datasets from field studies. Despite being developed for multi-Doppler lidar experiments, the methodology can be used for any field campaign. The majority of the presented steps are relevant for all field experiments (e.g., defining scientific question, planning infrastructure, etc.), while some are WindScanner specific (e.g., scanning mode design). In this manuscript, we presented a sequential execution of the steps. However, some steps can and should run in parallel (e.g., data archiving and dissemination with execution and data collection) and some steps can be merged together (e.g., experiment layout design and infrastructure planning).*

In the same section the last paragraph has been rephrased and updated with the latest information related to the methodology application beyond Perdigão-2015:

Original paragraph: *Despite the environmental conditions, challenges imposed by the site, and technical issues with the instruments, high-quality observations of the various flow aspects of the Perdigão site have been acquired, with a few highlights outlined in this paper. Also, the datasets are used for the design of a longer, and instrumentation-wise more extensive Perdigão-2017 field campaign, that will take place in the first half of 2017 over a period of six months.*

Revised paragraph: *Despite the environmental conditions, challenges imposed by the site, and technical issues with the instruments, high-quality observations of the various flow aspects of the Perdigão site have been acquired, with a few highlights outlined in this paper. Also, the datasets and presented methodology have been used for the design of a longer, and instrumentation-wise more extensive Perdigão-2017 field campaign, which took place in the first half of 2017 over a period of six months (Witze, 2017; Fernando et al., 2017). Besides the Perdigão-2015 and Perdigão-2017, the methodology has been successfully applied in several experiments, such as for example the Kassel-2016 and RUNE campaigns (Floors et al., 2016).*

On behalf of all co-authors,

Nikola Vasiljević and José Palma